# Assessing the Nationwide Benefits of Vehicle–Grid Integration during Distribution Network Planning and Power System Dispatching

Giuliano Rancilio [1],*, Alessia Cortazzi [2], Giacomo Viganò [3] and Filippo Bovera [1]

1    Department of Energy, Politecnico di Milano, Via Lambruschini 4, 20156 Milano, Italy
2    Centro Elettrotecnico Sperimentale Italiano (CESI), Via Raffaele Rubattino 54, 20134 Milano, Italy
3    Ricerca Sul Sistema Energetico (RSE), Via Raffaele Rubattino 54, 20134 Milano, Italy
*    Correspondence: giuliano.rancilio@polimi.it

**Abstract:** The diffusion of electric vehicles is fundamental for transport sector decarbonization. However, a major concern about electric vehicles is their compatibility with power grids. Adopting a whole-power-system approach, this work presents a comprehensive analysis of the impacts and benefits of electric vehicles' diffusion on a national power system, i.e., Italy. Demand and flexibility profiles are estimated with a detailed review of the literature on the topic, allowing us to put forward reliable charging profiles and the resulting flexibility, compatible with the Italian regulatory framework. Distribution network planning and power system dispatching are handled with dedicated models, while the uncertainty associated with EV charging behavior is managed with a Monte Carlo approach. The novelty of this study is considering a nationwide context, considering both transmission and distribution systems, and proposing a set of policies suitable for enabling flexibility provision. The results show that the power and energy demand created by the spread of EVs will have localized impacts on power and voltage limits of the distribution network, while the consequences for transmission grids and dispatching will be negligible. In 2030 scenarios, smart charging reduces grid elements' violations (−23%, −100%), dispatching costs (−43%), and RES curtailment (−50%).

**Keywords:** smart charging; vehicle–grid integration; distribution planning; power system dispatching; flexibility; reserve margins

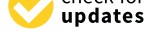



## 1. Introduction

Recent research highlighted that the transport sector contributes 16% of the overall worldwide $CO_2$ emissions [1]. Road transport's contribution is the largest one (12%), with 60% of its emissions arising from passenger travel, including cars, motorcycles, and buses [2]. In 2019, the transport sector was responsible for 25% of the total greenhouse gas (GHG) emissions in Italy, with the road sector contributing 92% of that portion [3]. In addition, fossil fueled transport generates a considerable volume of local pollutants, including $NO_X$, VOC, PM, and CO, whose adverse effects on human health are most worrying when they are emitted near populated zones, such as those related to road transport [4].

Considering the share of fossil fuel utilization in the road transport sector, and its contribution to the total GHG emissions in developed countries, vehicle decarbonization stands as one of the most efficient and effective ways to achieve the net-zero target fixed by the European Union in its Fit For 55 package [5]. This is also desirable due to the very poor energy efficiency characterizing internal combustion engines (ICEs) running road vehicles, which, during driving conditions, ranges between 20% and 30% [6]. As such, replacing ICE-based engines with fully electrified vehicles entails an energy efficiency increase of around 300%: this means that it will be possible to move the same vehicles along the same routes with 1/3 of the currently used energy [7,8]. For the case of Italy,

even considering the current energy mix for power production, this will generate a 50% reduction in the GHG emissions of the road transport sector [9]. Direct electrification as the main path for road transport decarbonization is supported by evidence on biofuels' and synthetic fuels' large-scale adoption. The main issues are related to the irremediable lower efficiency of ICEs with respect to electric motors [7], the technological immaturity of these solutions [10,11], the limited availability of fuels, and their potential competition in terms of land usage with other sectors, such as agriculture [12,13]. Also, ICE-based solutions often fail to prevent the emission of local pollutants [14,15].

Nonetheless, while direct electrification of road transport is the favored solution for reaching net-zero emissions in the transport sector, several experts, researchers, and politicians are raising awareness about its impact on the power grid, including both power system management (so-called dispatching) and electricity network development. Indeed, the concept of a power grid hosting capacity has been introduced and discussed over many years, indicating the amount (expressed in MW) of distributed power generation that a distribution network (DN) can host. Nowadays, the proliferation of dispersed generators, mainly using photovoltaic technology, together with the electrification of both transport and heating sectors with the diffusion of EVs and heat pumps, is placing even more pressure on the power system and creating troubles related to its adequacy, security, development, and management.

In [16], 20 municipalities in Portugal are considered under different EV diffusion scenarios for 2030. The authors use a Monte Carlo approach to show how a 20% penetration target for BEV diffusion in the circulating fleet of 20 municipalities would negatively impact the distribution grid of 35% of them in 2026 and 95% of them in 2030. Several research studies, such as [17,18], highlight how the impact of EV charging on peak demand is relevant even for low EV adoption rates, thus calling for an upgrade of the DN. The most important issues emerging concern feeders' and transformers' congestion levels, together with network voltage profiles. As concluded also by ENTSO-e [19], while several power issues are expected due to EV charging, especially concerning low-voltage grids, no significant challenges are forecasted in terms of overall energy consumption; indeed, EVs' contribution to the final electricity consumption will be between 1% and 6% in 2030 in all major countries according to the IEA [20]. Certain country-wide studies focused on Germany, both considering the impacts at the local and system levels [21,22], show that a small increase can be expected in overall peak energy demand due to EV penetration.

Besides being a potential source of challenges, EV charging could offer an opportunity for the power system. This aspect has been largely discussed in the literature, where different solutions for vehicle–grid integration (VGI) have been proposed, including smart charging (V1G and V2G) [23–26], EV charging infrastructures integrating a battery energy storage system (BESS) [27,28], and coordinated deployment between distributed energy resources and EV charging points [29–32]. The most common opportunities arising from EV charging management, as proposed and discussed in the literature, are the reshaping of the load charging curve, typically from peak (evening) hours to off-peak hours, which can help in reducing both TN congestion and DN overloads [33,34]; the provision of balancing services by EVs implemented in virtual power plants (VPPs) trading on ancillary service markets (ASMs), including reserve margin provision, frequency-response services, and replacement reserves [35–37]; the reduction of non-programmable RES (NP-RES) overgeneration, achieved by scheduling EV charging according to the forecasted overproduction of energy in a given zone [38]; the increase in energy self-consumption when EV charging points are placed behind the meter, with a corresponding reduction in the final user energy bill [39,40]. To the best of the authors' knowledge, these possibilities have been discussed in the literature based on real data or on analyses of specific case studies, while no literature review has been carried out, for instance, on charging behavior or how to use the research methodologies for nationwide studies.

This work intends to build on the past literature to provide a comprehensive but detailed analysis of both the impact of EV charging on the power system and the benefits of

the full exploitation of VGI solutions. We do so by considering the case of the Italian power system scenario for 2030 according to the most recently published energy and climate plans. The main novelties of this study can be summarized in three elements. First, all available on-grid VGI solutions are considered, thus including smart charging, V1G, V2G, BESS, and NP-RES integration. This allows us to understand the effectiveness and interaction of diverse VGI solutions with respect to the different impacts that the EV charging process has on the power system. Off-grid VGI solutions, including on-demand EV charging and battery swapping, are not considered since they are out of scope of this study. Second, VGI implementation is considered for a well-defined and formalized set of charging modes that covers all possible road transport vehicles, including people and goods transportation. Moreover, as will be explained, all EV charging management processes follow a *no-harm* approach: developed models include the need to avoid any impact on the quality and comfort perceived by the final users when it comes to the charging service. This potentially limits the exploitable flexibility but increases the acceptability and diffusion of VGI schemes. Third, this study considers both the local and system levels, providing a complete overview of the impacts and benefits for the power system at the distribution and transmission network levels (thus, broadening [22]). These three elements finally give insights into the economic convenience of EVs' flexibility at a country-wide scale, without focusing on a single technology, use case, or system operations need, but considering the overall benefit of VGI solutions for both system operators and EV users.

While this study takes the Italian system scenario in 2030 as a case study, both the implemented methodology and the obtained results are fully generalizable. This is because the technological solutions implemented, the charging modes and vehicle types considered, and the characteristics of the power network are qualitatively independent from the selected national scenarios. This is demonstrated by the fact that archetypes from the international literature are utilized here for both charging behaviors and power network characteristics.

The remainder of this paper is organized as follows. Section 2 introduces the reference modeling scenario, including the power system consistency, forecasts about the future diffusion of EVs, and a framework for classifying and characterizing the different EV charging modes considered. Then, it describes the methodology used to define the EV charging profiles and impacts on both DN and TN. Section 3 presents the results obtained, including the charging schedules, together with the calculated flexibility profiles, the impact of dumb charging on both DN and TN, and the benefits of VGI solutions for the overall power system. Section 4 discusses the results, presenting the main implications and listing possible future improvements.

## 2. Materials and Methods

### 2.1. Reference Modeling Scenario and Main Assumptions

2.1.1. Power System Scenario According to Italian 2030 Plans

After extensive work carried out at the European level, the Clean Energy for all European Package (CEP) was finalized and adopted in 2019, marking significant progress made with respect to the energy union strategy published in 2015 [41] and drawing the EU climate and energy strategy towards 2030. All CEP actions, grouped in a climate and an energy component [42], are under the umbrella of the Governance Regulation (2018/1999), which requires Member States to prepare a National Energy and Climate Plan (NECP) and a Long-Term Strategy (LTS) to clearly state their intervention program designed to reach 2030 and 2050 targets. Italy published its NECP in January 2020 [43].

In December 2019, the European Commission set out a European Green Deal aiming at a carbon-neutral continent by 2050, with an intermediate step of −55% GHG emissions by 2030 (compared to 1990) [44]. The European Climate Law, presented in March 2020, was the first act of this journey [45]; however, its announcement, amendment process, and entering into force coincided with the widespread outbreak of the COVID-19 pandemic, which inevitably had an influence. Following the peak of the pandemic, the EU Commission presented in July 2021 a new package of proposals, called the Fit For 55 (FF55) package [5].

This foresees a set of policy instruments tackling the green transition with a cross-sectoral approach, including carbon pricing systems, efficiency and sustainability targets, energy and environmental binding rules, and financing tools for supporting a just transition.

The definition of new targets for carbon emission reduction by 2030 calls Member States to update their NECPs; indeed, the updated version of the Italian NECP was published in 2023. Following this deadline, forecasts of the Italian scenario in 2030 were published by different parties, including the Italian power and gas TSOs [46] and the confederation of Italian industries [47]. In this paper, we consider an Italian power system scenario coherent with the FF55 policies, including 65% of the final electricity demand covered by renewable energy sources (RESs). In detail, the most important assumptions concern the installed generation and storage capacity, the national power balance, and the electricity market prices.

First, the reference Italian scenario for 2030 considers about 130 GW of RES generation capacity installed, including 75 GW of photovoltaic and 27 GW of wind plants. The installed natural gas generation amounts to 46 GW, while coal-based production is absent, coherent with the Italian NECP published in 2019. Beside this, the overall volume of new storage capacity considered is equal to 15 GW, with an average energy-to-power ratio slightly above 6 h. The largest share of storage represents utility-scale batteries (11 GW), while the remainder constitutes distributed small batteries. Second, total electricity demand is between 362 and 366 TWh/y, varying according to the EV penetration scenario considered (see Section 2.1.2). This demand is largely concentrated in the north market zone (54%) and corresponds to a national yearly energy production of 322 TWh, divided into 244 TWh coming from RES and 81 TWh from conventional generation. The remainder is imported from foreign bidding zones. Third, simulations of the day-ahead electricity market (DAM) show a mean price of around 110 EUR/MWh for those market zones with the highest demand (north, center-north, and center-south), while a lower price level (around 100 EUR/MWh) results for the other market zones (south, Calabria, Sicilia, and Sardegna). These average prices have been used as the basis for the price profiles, which have hourly values and were built considering the variability of power generation and demand and the corresponding volatility of market prices. Day-ahead simulations of the power system highlighted a residual RES overgeneration of 2.1 TWh/y because of the DAM unit commitment. Figure 1 graphically shows the installed generation capacity and the annual energy balance considered for Italy for 2030.

2.1.2. EV Circulating Fleet in Italy in 2030: Composition and Characteristics

This study considers two possible scenarios concerning the number and the consistency of EVs in the circulating fleet in Italy in 2030. The base-case scenario is consistent with the Italian NECP policies published in 2019, while the accelerated scenario is coherent with FF55 targets. These two scenarios (base case and accelerated) differ for the penetration of four different categories of EVs: cars, light commercial vehicles (LCV), heavy commercial vehicles (HCV), and public transportation (PT). Cars include both battery-only (BEV) and plug-in hybrid (PHEV) vehicles.

First, cars are by far the most abundant category of EV considered. The total number of cars considered is 6 million in the base-case scenario and 7.5 million in the accelerated one. The BEV share is 85% in the accelerated scenario, while it is 66% in the base case. Based on a previous analysis, it was possible to estimate the territorial diffusion of electric cars, forecasting their penetration at the municipality, province, and region levels based on a combination of four main parameters: the current number of circulating EVs, the per capita income, the air quality, and the availability of private garages. The local availability of garages was considered a major influence on EV diffusion, considering the importance of nighttime charging. Furthermore, appropriate weight was given to air quality, considering the diffusion of municipal restrictions on the circulation of polluting vehicles. Finally, the current diffusion of EVs and the average wage within a territory were considered positive factors promoting the adoption of EVs. As reported in Figure 2, this evaluation procedure

allowed us to compute the numbers of electric cars circulating in two different spatial configurations: considering the seven Italian market bidding zones, and distinguishing urban and rural areas. In this latter classification, 44 urban areas were flagged as municipalities with more than 100,000 inhabitants, among which 59% were in the northern part of Italy.

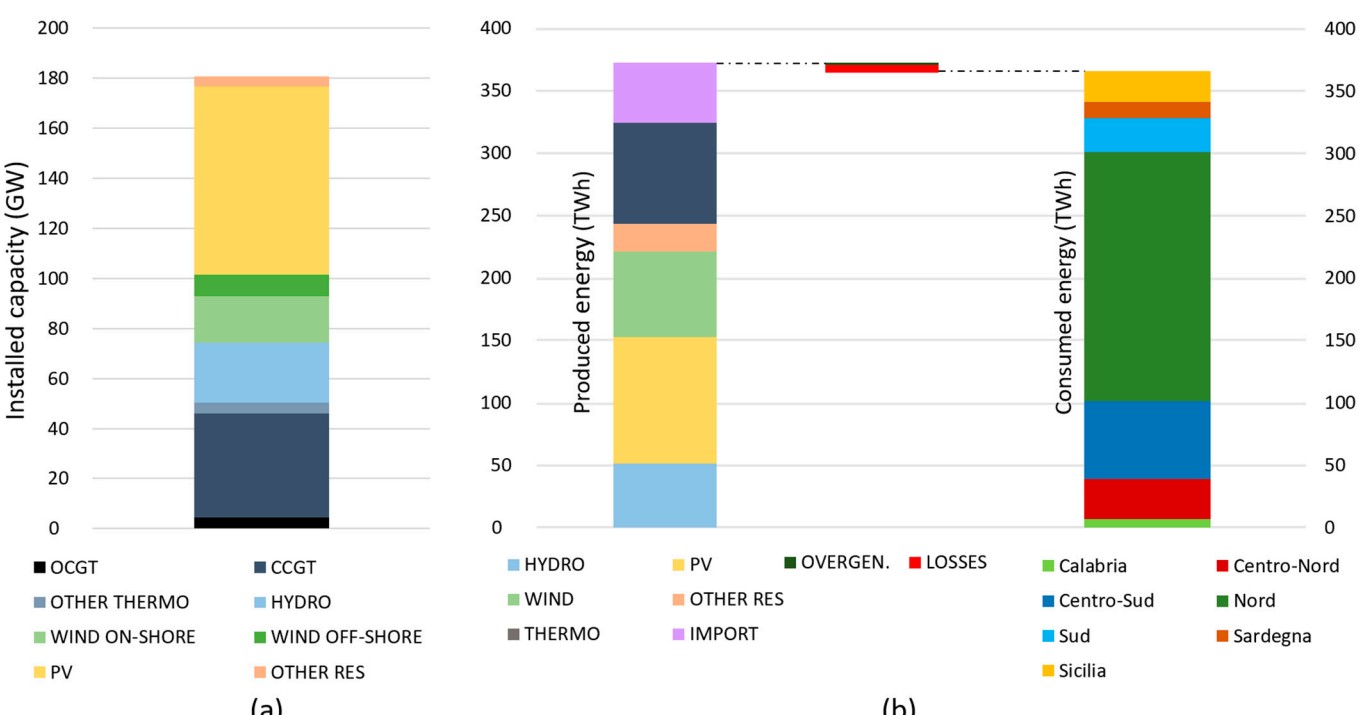

**Figure 1.** Reference scenario of installed capacity in GW (**a**) and yearly national energy balance in TWh (**b**) for Italian power system in 2030. Consumed energy equals produced energy, minus losses and overgeneration.

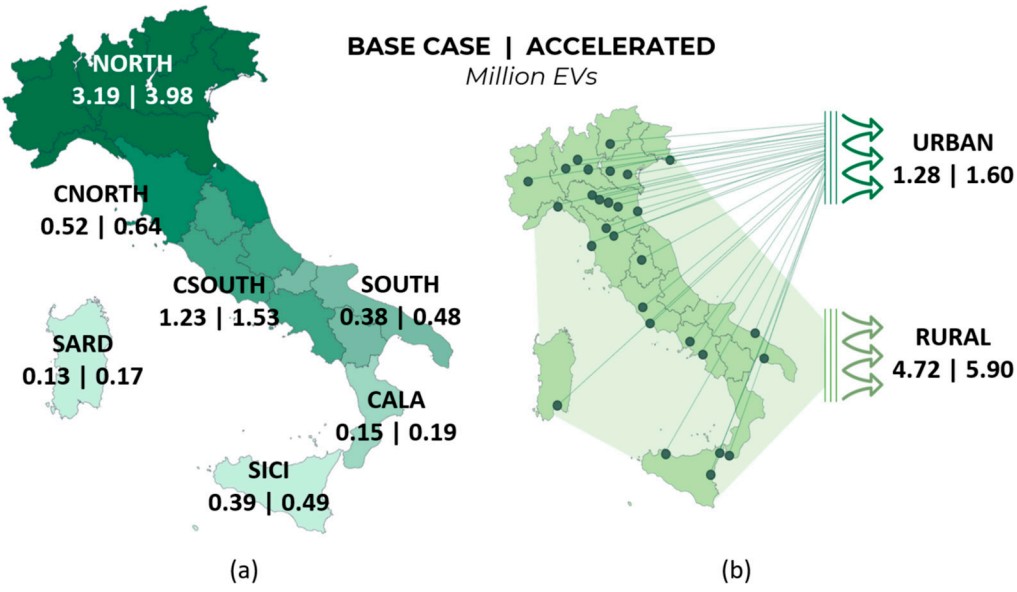

**Figure 2.** Numerosity and distribution in the Italian territory of electric passenger cars in 2030, according to base-case and accelerated scenarios, clustered according to the electricity market bidding zones (**a**) and population density (**b**). Each zone is highlighted with a different color (on the left). Dark dots show the cities considered in the urban diffusion, while the rest of the territory (light green) is considered in rural cluster.

Beside the numerosity and territorial distribution of electric cars, a further hypothesis was formulated about vehicles' segmentation and characteristics. Data available from Italian [48] and European [49] databases, together with ad hoc research activities conducted on electric cars, were firstly used to classify and characterize different segments of electric cars. According to EU nomenclature [50], BEV cars were divided into segment A (mini and city cars), segment B (small cars), segment C (lower medium cars), segment D (large cars), and segment E+ (executive, luxury, and sport cars). For PHEV cars, only a distinction was made between small cars, including segments A–C, and large cars, including segments D and E+. Based on this classification, all BEV and PHEV models currently available were classified, allowing for an analysis of the average battery size and charging power for each segment. Table 1 presents the resulting assumptions used within the subsequent analysis. Regarding the capacity of the battery, in 2030, an increase of +10% with respect to current values was considered, to include technological evolution. Two values were considered for the charging power, distinguishing between AC and DC charging modes. Finally, data on circulating Italian cars' segmentation in 2021 [48] were used to calculate the assumed weight of each segment forming part of the overall number of electric cars assumed in the base-case and accelerated scenarios.

**Table 1.** Passenger car reference data and shares used in the analysis, distinguishing between plug-in hybrid and battery electric vehicles.

| Category | Battery (kWh) | AC Charging Power (kW) | DC Charging Power (kW) | Share of Total |
|---|---|---|---|---|
| PHEV A-B-C | 12 | 3.7 | - | 14% |
| PHEV D-E+ | 15 | 7 | - | 2% |
| BEV A | 45 | 7 | 50 | 35% |
| BEV B | 55 | 11 | 50 | 15% |
| BEV C | 65 | 11 | 100 | 22% |
| BEV D | 80 | 11 | 100 | 8% |
| BEV E+ | 100 | 11 | 150 | 4% |

Similar considerations were developed for goods and public transport, considering their behavior and battery capacity. Again, data from refs. [48,49] were utilized to understand the territorial distribution of LCV, HCV, and PT, where the LCV category was associated with vehicles weighting less than 3.5 tons. Table 2 presents all the main assumptions in terms of vehicles' diffusion and characterization.

**Table 2.** Main assumptions and reference data concerning goods and public transportation used in the analysis.

| Category | Base Scenario Diffusion (kEV) | Accelerated Scenario Diffusion (kEV) | Battery (kWh) | AC Charging Power (kW) | DC Charging Power (kW) | Consumption (kWh/100 km) | Mileage (km/year) |
|---|---|---|---|---|---|---|---|
| LCV | 530 | 750 | 75 | 22 | 150 | 35 | 20,000 |
| HCV | 30 | 50 | 400 | 22 | 350 | 150 | 35,000 |
| PT | 5 | 7 | 460 | 22 | 350 | 150 | 45,000 |

### 2.1.3. Classification of EV Charging Modes

The link between EVs and power systems occurs when charging. A set of EV charging modes were studied: these are representative situations (and conditions) that characterize the EV charging processes. Several works already discussed this issue [51–57], drawing up useful classification frameworks, yet those were often supported by datasets that were limited in time and space. For the scope of this study, a full framework for clustering the diverse EV charging modes is proposed, building on the available literature and specific conversations with several operators in the EV sector, including vehicle manufacturers, charging point operators (CPOs), mobility service providers (MSPs), and other stakeholders.

As reported in Table 3, six charging modes are proposed for electric cars, while three charging modes are presented for goods and public transportation. The most important assumptions concern the parking duration and the maximum power available at the charging station. Specific hypotheses for every charging mode have been introduced based on a set of parameters, including the arrival and departure time; the battery state-of-charge (SoC) at the beginning of the charging process, and the target one to be reached before the vehicle is picked up by its user; and the assumed penetration of vehicle-to-grid (V2G) technologies allowing a bi-directional power flow from and to the car. For all these variables and for each charging mode, a specific probability distribution around a mean value is considered, to develop charging and flexibility profiles for all charging modes. This procedure is described in detail in the next sections. For now, we highlight that the proposed SoC range influences the requested energy per charging event. Thus, the possibility of incomplete charging is considered. This results in more charging events for the same overall energy demand.

**Table 3.** Definition and characterization of EV charging modes, with reference values utilized for state-of-charge management, charging power, and V2G penetration.

| Icon | Charging Mode | Stop Duration | Range of Initial–Final SoC (%) | EVSE Charging Power (kW) | V2G Penetration |
|---|---|---|---|---|---|
| | Residential | Long (>10 h) | 30–60/80–100 | 3–6 | 0% |
| | Workplace | Employees: 8 h Fleet: >10 h | 30–70/80–100 | 7–22 | 30% |
| | Public—slow | Medium (3 h) | 10–40/50–80 | 22–50 | 20% |
| | Public—fast | Stop and go (<<1 h) | 20–50/50–80 | 50–300 | 0% |
| | B2C—shopping center | Short (1 h) | 30–70/40–100 | 22–50 | 20% |
| | B2C—interchange parking | Long (6 h) | 30–70/100–100 | 7–22 | 20% |
| | Light commercial vehicle (LCV) | Medium (4 h) | 40–70/60–100 | 22–150 | 50% |
| | Heavy commercial vehicle (HCV) | Long (6 h) | 10–40/80–100 | 22–150 | 50% |
| | Public transport (PT) | Long (6 h) | 10–20/90–100 | 22–150 | 50% |

While LCV, HCV, and PT charging modes are the only alternatives for commercial and public vehicles, passenger cars' charging modes have different weights in each typical day and geographical context. This means that the overall charged energy in a specific area on each typical day is shared among different charging modes, as presented in Figure 3. The weights were selected based on an extended literature review of institutional, industrial, and academic sources [51–57]. Generally, residential charging represents 50% of charged energy and it has a larger penetration than average during holidays and in rural contexts. Workplace charging occurs differently, largely during workdays. Public charging has a larger role than average in metropolitan contexts, partially substituting residential charging (since there are fewer private garages). B2C charging has a non-negligible penetration, especially at shopping centers during weekends and holidays.

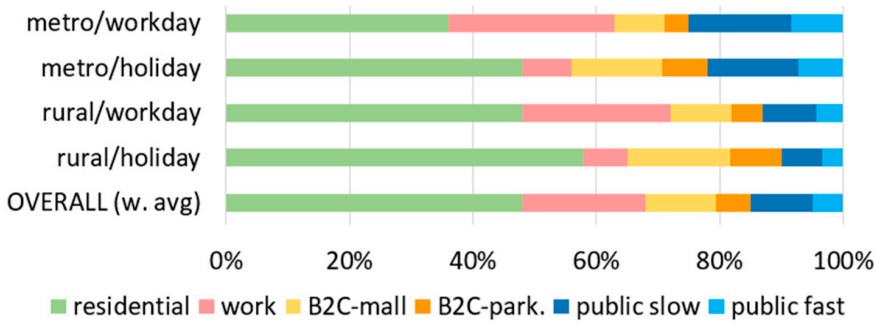

**Figure 3.** Charging mode breakdown for passenger cars in different contexts (metropolitan and rural) and typical days (workday and holiday).

*2.2. Proposed Methodology*

2.2.1. Definition of EV Charging Profiles and Flexibility Available

The goal was to estimate the charging and flexibility profiles for each charging mode and for each Italian electricity market zone. To do so, the first step consisted of defining a presence profile for each charging context, which was then transformed into an entry and exit profile for each charging mode. A wide literature review was performed, updating and extending a previously developed study [57] to also include LCV, HCV, and PT. The assumption was that the presence profile coincided with the connection period of the EV to the EV supply equipment (EVSE). For example, the presence profile for a household [55] was considered equal to the time period during which the EV was connected to the home wallbox, and similarly, the entry/exit profile for employees and customers of a shopping center [58] became the start/end of the EV connection. After processing the sources [55,57–61], the analysis returned an entry and exit hourly profile for 24 h and for each charging mode. Where significant differences were present, separate profiles for the working days and for holidays/weekends were proposed.

The entry and exit profiles were used as inputs for a Monte Carlo procedure, as well as the data in Table 3. Two separate Monte Carlo procedures were carried out, for cars and for other vehicles. The first procedure used the data in Table 1, while the second was based on Table 2. The Monte Carlo procedure generated N (=1000) EV objects, characterized by the quantities listed in Table 4.

**Table 4.** Metadata associated with each EV object within the Monte Carlo procedure.

| Key | Symbol | Unit of Measure | Notes |
|---|---|---|---|
| Battery capacity | $C_i$ | kWh | Based on the selected EV segment |
| Charging power | $P_i$ | kW | Minimum between the EVSE charging power (based on charging mode) and the EV charging power (based on EV segment) |
| V2G flag | - | - | Boolean based on the V2G penetration of the considered charging mode |
| Entry time | $T_{in,i}$ | h | The connection time of the EV, based on a random extraction from the entry profile of the considered charging mode |
| Exit time | $T_{fin,i}$ | h | The disconnection time of the EV, based on a random extraction from the exit profile of the considered charging mode |
| Stop duration | $\Delta T_i$ | h | $= T_{fin,i} - T_{in,i}$ |
| Initial SoC | $SoC_{in,i}$ | % | Based on the SoC distribution for the considered charging mode |
| Desired final SoC | $SoC_{target,i}$ | % | Based on the SoC distribution for the considered charging mode |
| Requested energy | $E_i$ | kWh | $= \left( SoC_{target,i} - SoC_{in,i} \right) \times C_i$ |
| Real final SoC | $SoC_{fin,i}$ | % | $= \min \left( SoC_{target,i}, SoC_{in,i} + \Delta T_i \times P_i \right)$ |

Each of the N EV starts charging at its entry time ($T_{in,i}$). The charging power ($P_i$) is the minimum between EVSE and EV nominal power. The end of charging for the i-th EV ($EoC_i$) is computed as follows.

$$\begin{cases} EoC_i = T_{in,i} + \frac{E_i}{P_i} & if\ SoC_{fin,i} = SoC_{target,i}\ and\ T_{in,i} + \frac{E_i}{P_i} < 24 \\ EoC_i = T_{in,i} + \frac{E_i}{P_i} - 24 & if\ SoC_{fin,i} = SoC_{target,i}\ and\ T_{in,i} + \frac{E_i}{P_i} \geq 24 \\ EoC_i = T_{fin,i} & elsewhere \end{cases} \quad (1)$$

Indeed, the $EoC_i$ could be equal to the exit time ($T_{fin,i}$) if the desired SoC ($SoC_{target,i}$) is only obtained while charging at full power for the whole stop duration ($\Delta T_i$) or is not obtained. Otherwise, the $EoC_i$ occurs between the entry and exit times. To account for the stops that last overnight, $T_{in,i}$ can be larger than $T_{fin,i}$. Similarly, $T_{in,i}$ can be larger than $EoC_i$, indicating that charging ends the day after entry (this simplification only works in the assumption of a persistence model based on typical days). Following the procedure for each EV, Boolean arrays indicating the *Presence, Charging,* and *NOT Charging* periods are retrieved, as illustrated in Figure 4 for different situations. As can be seen, if the stop duration is longer than the time needed to reach $SoC_{target,i}$, the *Presence* and *Charging* arrays are different. The difference between the *Presence* and *Charging* arrays defines the *NOT Charging* array. The *Power* array reports the charging power, which is generally equal to $P_i$. Only during last charging hour is the power equal to the minimum between $P_i$ and the average power to reach $SoC_{target,i}$ at the end of the hour.

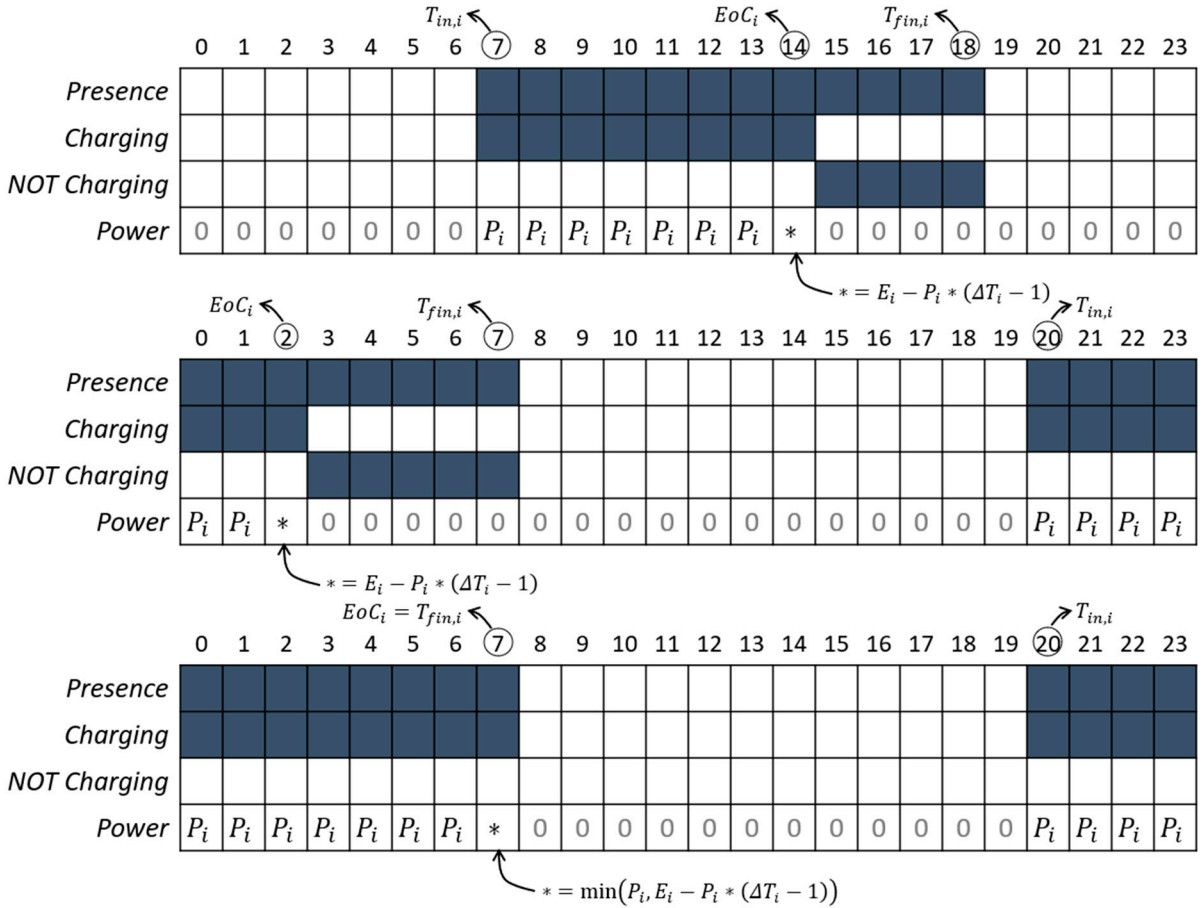

**Figure 4.** The 24 h Boolean arrays for each EV object in the Presence, Charging, and NOT Charging periods (blue cells = 1; white cells = 0), and numerical arrays for charging Power.

For each charging mode, *Power* arrays for the N EVs are summed for each charging profile, each typical day, and each geographical context. Four typical days are considered:

weekday winter, weekday summer, holiday winter, and holiday summer. Two geographical contexts are considered, specifically "metropolitan area" and "other areas". This considers that the commuting patterns in large cities are significantly different from sub-urban and rural areas due to a set of factors, including the larger diffusion of public transport and the lower number of private garages [8]. Once a power profile has been reconstructed for each typical day and context (eight profiles) and for each charging mode (nine modes, as seen in Table 3), it is rescaled for each market zone. This is to obtain a yearly power profile for each charging mode and market zone.

It must be recalled that each charging mode is responsible for a share of the total charged energy ($r_\%$). This share depends on the geographic area (rural vs. metropolitan) and on the typical day (work or holiday) (see Figure 3 presenting the breakdown of charging). The total energy demand for each charging mode on each typical day for EVs in each zone is computed as the integral of the power. The power profiles are obtained as in the next equation.

$$P_{i=0\ldots23,j,k,l,m} = p_{i=0\ldots23,j,k} \times \left( \frac{E_{j,k,l,m}}{\sum p_{i=0\ldots23,j,k} * \Delta t} \right) \tag{2}$$

$p_{i=0\ldots23,j,k}$ is the charging profile obtained as the outcome of the Monte Carlo procedure for each hour $i$, each charging mode $j$, and each typical day $k$. $E_{j,k,l,m}$ is the energy demand for the charging mode $j$, the typical day $k$, the geographical context (rural or metropolitan) $l$, and the zone $m$. $\Delta t$ is one hour in this study since we have hourly profiles as both inputs and outputs. We obtain as the output $P_{i=0\ldots23,j,k,l,m}$, which are the power profiles in MW for the EV demand in each context. Putting in series the power profiles $P_{i=0\ldots23,j,k,l,m}$, we can reconstruct the yearly power profile for a charging mode in each zone; summing the charging modes, we can reconstruct the zonal demand for EV charging; and summing the zones, we can determine the Italian demand.

Once we have all the charging profiles, we estimate flexibility. Flexibility is obtained in this study assuming a no-harm approach: each car can provide flexibility if its stop duration is sufficient to reach $SoC_{target,i}$ from $SoC_{in,i}$ before the end of stopping ($T_{fin,i}$). This is the case for the top and mid charts of Figure 4. In that case, the EV can decrease the power input to zero to provide upward flexibility (V1G) or even discharge the battery to provide more flexibility (V2G). In any case, $SoC_{fin,i}$ must be greater or equal to $SoC_{target,i}$. Both the energy and power margin must be computed to assess flexibility. The approach can be better understood from Figure 5 with an example considering a car stopping from 10 AM to 2 PM and reaching $SoC_{target,i}$ before $T_{fin,i}$ (we use load convention for the power sign). The charging power ($P_i$) is shown (top diagram) as well as the maximum power ($P_{max}$) that can be withdrawn to offer downward flexibility, and the minimum power ($P_{min}$) can be as low as 0 for V1G and negative (discharge) for V2G. The energy variation when charging ($E_{ch}$) is presented (bottom diagram), as well as energy flexibility with the shaded areas: the downward flexibility (blue) considers charging at $P_i$ even overpassing $SoC_{target,i}$, while upward flexibility considers the minimum energy content in the battery (minimum SoC) for reaching the $SoC_{target,i}$ at $T_{fin,i}$.

The flexibility profile in a market zone for a charging mode is obtained by summing the power and energy flexibilities for each EV with at least one element of the *NOT charging* array equal to 1 (see top and mid diagram in Figure 4). The result is yearly profiles with an hourly resolution (considering working and holidays, cold and warm seasons) summing up rural and metropolitan demand and flexibility for each of the nine charging modes and seven electricity market zones for Italy; these total 63 (=7 × 9) VPPs or aggregates of EVs bidding in the electricity markets. Charging profiles characterize the demand that EVs place on DAM. Flexibility power and energy profiles represent EVs' capability of providing balancing services on the ASM.

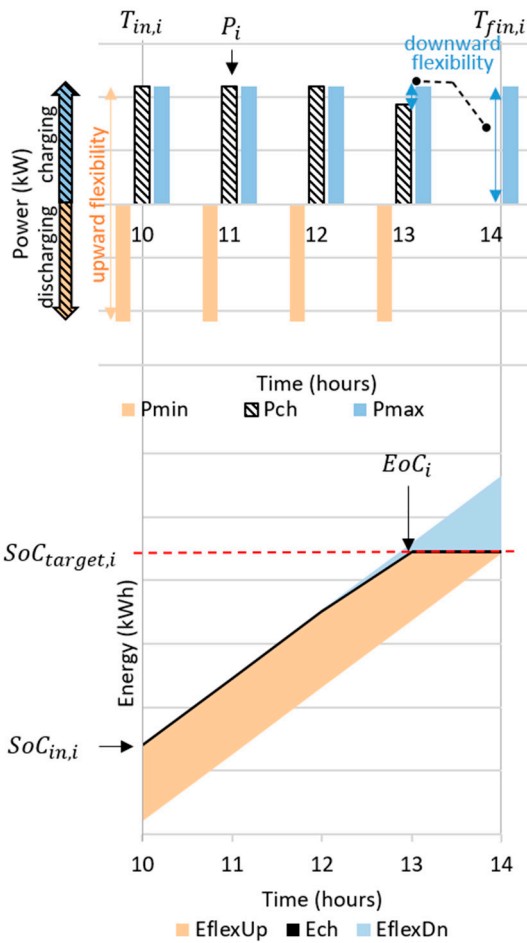

**Figure 5.** Power and energy flexibility estimation for a generic EV object.

2.2.2. Estimating the Impacts of EV Charging on Distribution Network Planning and Identifying Its Benefits

Two typical DNs were developed by using synthetic networks for the MV level and representative networks for the LV level. The aim of this procedure is to obtain the power and voltage profiles for all network components, including HV/MV and MV/LV transformers, and MV and LV lines; this will be achieved for different typical days and geographical contexts. MV networks with urban and rural characteristics were selected from a set of synthetic networks representing the Italian distribution system [9]. Their schemes are given in Figure 6, where the peculiar features of each area are visible: the urban network presents a larger demand concentration and most of the nodes are withdrawing power from the grid (green and blue dots); rural areas see larger distances and a wider penetration of DG (yellow to red dots are injecting power into the grid). In addition to the electric characteristics, there is information about loads and generators, such as the connection node, the nominal power, the electric user type (e.g., residential, tertiary, industrial) and generators (e.g., photovoltaic, cogenerator), and the relative hourly profiles for twelve representative days (i.e., working days, Saturdays, and Sundays for the four seasons).

The model of the LV networks was drawn from [10]. For each node of the MV network with a connected load, an MV/LV transformer and the corresponding LV networks are assigned. The size of the MV/LV transformer, and the type and number of LV feeders, were selected based on the total load, which was randomly distributed among the LV nodes. The characteristics of the resulting networks are reported in Table 5.

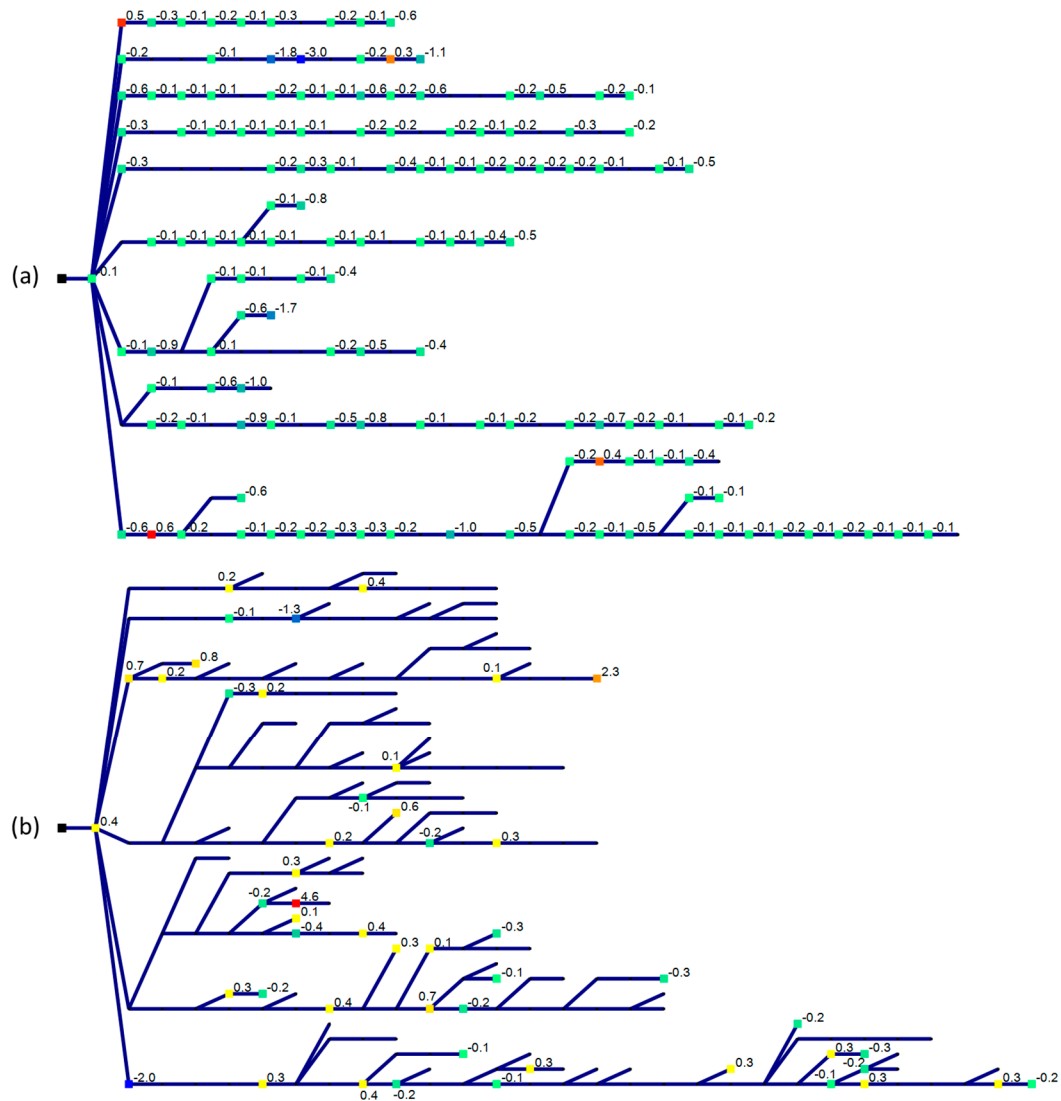

**Figure 6.** Schemes for the considered synthetic MV networks for urban (**a**) and rural (**b**) areas. The number and the color scale are coherent with the maximum exchanged power in MW (positive, yellow to red, for injection; negative, green to blue, for withdrawal) for the MV users (>0.1 MW of maximum exchanged power).

**Table 5.** Metadata of the adopted networks.

|  | Rural Network | Urban Network |
|---|---|---|
| Number of MV nodes [#] | 228 | 178 |
| Number of LV nodes [#] | 1371 | 2575 |
| MV lines' length [km] | 192 | 44 |
| LV lines' length [km] | 95 | 113 |
| HV/MV transformer power [MVA] | 25 | 63 |
| MV/LV transformers [#] | 105 | 103 |
| MV/LV transformers' power [MVA] | 17.4 | 47 |
| Maximum load [MW] | 10.2 | 37.2 |
| Maximum generation [MW] | 10.5 | 2.7 |

The charging stations and the served EVs were determined with the following steps. The number of EVs served by each network was computed distributing the EVs and the EV charging modes coherently with EV penetration scenarios, charging breakdowns, and the rural or urban share of EVs. The weight associated with each network, and therefore the

circulating EV number, was proportional to the nominal power of the HV/MV transformers relevant to the area. Based on the daily served EVs in the network, the number of charging stations was estimated considering a number of charging points per charging station and a number of served EVs per charging point, as reported in Table 6.

**Table 6.** EVSE data considered in simulations of the distribution networks.

|  | Residential | Work-place | Public Slow | Public Fast | B2C Mall | B2C Park | LCV | HCV | PT |
|---|---|---|---|---|---|---|---|---|---|
| Charging points per charging station | 1 | 5 | 4 | 6 | 8 | 8 | 4 | 6 | 10 |
| Daily vehicles per charging point | 1 | 1 | 10 | 10 | 10 | 1 | 4 | 3 | 1 |

The nominal power of each EVSE was assigned randomly based on the ranges shown in Table 3. The nominal power of each charging station was computed by summing the nominal power of its charging points: if the total power of the charging station was lower than 200 kW, it was connected to the LV nodes. The residential charging stations were distributed proportionally to the residential loads, and the other stations were distributed proportionally to the remainder of the load. The charging points were populated with EVs with a charging time distribution based on charging profiles obtained through the methodology previously described. Tables 7 and 8 provide an overall view of the considered network's size.

**Table 7.** Number of charging stations in tested networks.

| Network | Residential | Workplace | Public Slow | Public Fast | B2C Mall | B2C Park | LCV | HCV | PT |
|---|---|---|---|---|---|---|---|---|---|
| Urban | 1865 | 210 | 8 | 4 | 17 | 6 | 23 | 2 | 1 |
| Rural | 744 | 62 | 3 | 2 | 3 | 1 | 9 | 1 | 1 |

**Table 8.** Number of served EVs, overall energy supplied, and maximum withdrawn power in tested networks.

| Network | Cars | LCV | HCV | PT | Total Energy (MWh) | Max Power (MW) |
|---|---|---|---|---|---|---|
| Urban | 3885 | 363 | 24 | 8 | 35 | 4 |
| Rural | 1283 | 144 | 10 | 4 | 13 | 1.6 |

The outcomes of the procedure are the power and voltage profiles along the network components, simulated on twelve representative days, with hourly profiles. We exploited a dedicated power flow algorithm (Matpower (https://matpower.org/)) that allowed us to compute all the requested voltages and currents. The reference representative days include a winter's working day (WiW), Saturday (WiS), and holiday (WiH); a spring working day (SpW), Saturday (SpS), and holiday (SpH); a summer's working day (SuW), Saturday (SuS), and holiday (SuH); and an autumn working day (AuW), Saturday (AuS), and holiday (AuH). Finally, for the HV/MV transformers and MV lines, overloading is considered if the maximum current exceeds 60% of its nominal value; this also considers possible network reconfigurations. Instead, since usually there is no automation at the LV level, for MV/LV transformers and LV lines, the maximum current is set at 80% of its nominal values. The voltage limits for LV nodes are equal to ±10% of the nominal voltage.

Consequently, three different VGI solutions are simulated to evaluate possible benefits of EV charging for DN power and voltage profiles. First, it is assumed that connected EVs can modulate their power withdrawal, distributing their energy absorption along their stop duration and thus implementing so-called smart charging techniques. Smart charging is

implemented whenever possible on all charging modes but, in particular, in the workplace, B2C mall, B2C park, and residential parking. While in dumb charging, the maximum charging power $P_i$ is always assumed, when smart-charging solutions are implemented, the power absorbed is limited according to the EV user's needs and network constraints, thus exploiting the flexibility margins available, as introduced in Section 3.1. The smart-charging techniques implemented include both V1G and, whenever possible, V2G solutions. A second solution considers the installation of a battery to support fast-charging stations. A BESS acts as a low-pass filter, reducing power peaks, especially when a lot of charging points are operating along the same feeders. BESSs are installed at charging stations with a maximum charging power of between 50 and 200 kW; their nominal power is equal to that of the station they are connected to, and their energy capacity ensures a maximum charging/discharging duration of one hour at the nominal power (energy-to-power ratio of 1 h). In detail, the adopted control strategy assumes that the BESS is fully charged (SoC of 100%) at the beginning of the simulated day; then, it discharges to keep the maximum power withdrawal from the grid below a specific threshold, which has been fixed at 30% of the EVSE nominal power. When the requested charging power returns below the specified threshold, the BESS starts charging in order to be fully charged by the end of the day. Finally, a third solution foresees the installation of photovoltaic (PV) plants near certain loads connected to the DN. A 3.3 kW PV generator is assigned to all the non-domestic charging stations: this is expected to increase possible synergies between daily EV charging and PV generation by lowering the distance between power production and consumption. The costs of BESS and PV systems are not considered since the objective of the analysis is to show only their technical benefit; otherwise, a more complex and comprehensive model should be used that integrates other services (e.g., arbitrage). It is important to note that implemented smart-charging solutions are not optimized. Specifically, installed PV generators and batteries are equally distributed over all charging points, allowing us to assess whether a non-optimal implementation of these solutions can improve network performances when there is a lack of coordination between CPOs and DSOs.

2.2.3. Evaluating the Impacts of EV Charging on Power System Dispatching and Identifying Its Benefits

In the reference case, EVs behave as loads, where charging profiles are estimated as described in Section 3.1 and EVs are not able to provide flexibility since they do not participate in the ASM. In this study, the impact of EVs on power system dispatching was modeled thanks to MODIS (Market Operation and DISpatching), a multi-area market simulator developed by the CESI to simulate the Italian ASM [11]. Additionally, the DAM dispatching was simulated using CESI-owned market simulator PROMEDGRID [62], whose outputs are a day-ahead scheduled program of generating units, forecasted RES generation, zonal electricity demand, inter-zonal power exchanges, and market prices at the zonal level. MODIS acquires the results of the DAM simulation and performs an hourly dispatching optimization over a target year, while respecting the main operating and security constraints. These security constraints, including the necessary reserve margins, are imposed by the system operator (SO) network code and the relevant equations are considered. The tool reproduces the balancing actions on the diverse units along the market sessions of the Italian ASM, including the ex ante (ASM ex ante) and balancing market (BM) phases (see Figure 7).

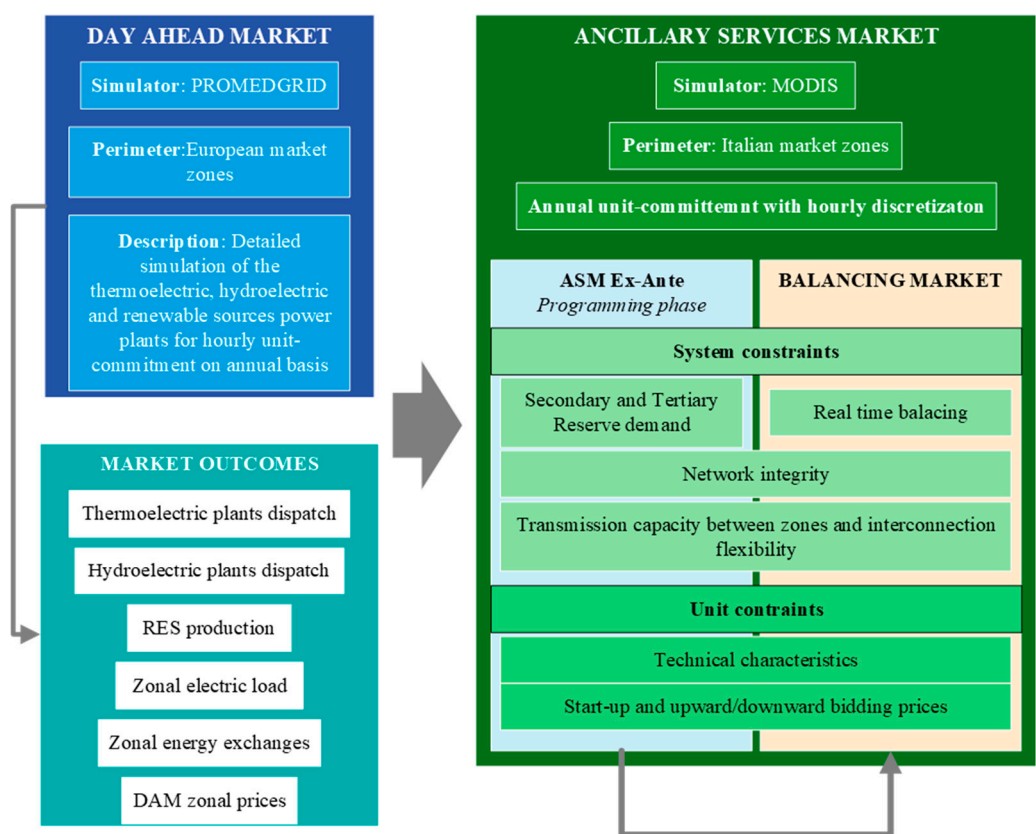

**Figure 7.** Block diagram of the market models used in the power system dispatching simulation.

This security-constrained unit commitment (SCUC) problem is solved using a deterministic optimization approach, aiming at minimizing the total costs sustained by the SO to balance the power system and keep it secure. Such a problem is formulated as mixed-integer linear programming (MILP), exploiting Gurobi as the solver (https://www.gurobi.com/). The most important constraints are as follows:

- Power flow constraints between different market zones based on the net transfer capacity (NTC) model;
- Power balance equations, taking into account power imbalances linked to forecast errors of NP-RES generation and load;
- Frequency restoration reserve (automatic and manual FRR) and replacement reserve (RR) needed to operate the system under safety constraints;
- Network integrity constraints, identifying the need for a certain number of production units to be operational in order to ensure the national power system's safety according to TERNA. These constraints are normally defined by geographical clusters, grouping plants that influence a given network branch;
- Units' technical constraints, such as minimum start-up and shutdown times, minimum and maximum available capacity, start-up and setup change requirements for thermoelectric units, and state-of-charge constraints for limited-energy-content resources;
- Upward and downward regulation bids presented on the ASM by all participating units, considering a pay-as-bid market mechanism, as is currently applied in Italy;
- Technical constraint units with a limited energy content.

Market models take as their input the Italian power system scenario in 2030, as previously introduced; this includes 10 market bidding zones, 15 inter-zonal interconnections, and more than 150 generation units or aggregates of units, including thermal, hydroelectric, electrochemical storage, demand response, and renewable-based units.

Power system dispatching, and thus the impact of EV charging on power system operations, is assessed as follows.

- DAM hourly schedules for each unit or aggregate of units are considered to assess the available power reserve margins. At the same time, the power reserves needed for both FRR and RR are calculated according to the Italian grid code [63]; if the available margins are lower than the necessary reserves, DAM schedules of units participating in the ASM are modified accordingly during the so-called ex ante phase.
- During the balancing phase of the ASM, units' schedules are further adjusted to ensure the real-time balance of the power system, coping with the imbalances caused by demand and NP-RES forecast errors.
- Both the DAM and ASM results imply the need for NP-RES curtailment, meaning that not all RES production can be hosted in the power system, mainly because of power reserve constraints combined with network capacity limits.
- Power system dispatching costs and technical impacts are hence evaluated considering the statuses of the power system before and after the ASM, thus considering all the actions taken by the SO to maintain the power system's security based on the provided boundary conditions.
- Environmental impacts are also calculated considering the negative externalities of pollutant emissions on society: the social cost, expressed in EUR/ton, represents the total net damage to the society of an extra ton of a specific pollutant. The substances considered include $CO_2$ and gaseous ($NO_X$, $SO_2$) and particulate ($PM_{2.5}$, $PM_{10}$) pollutants. Concerning $CO_2$ monetization, since the cost of Emission Trading Scheme (ETS) permits was already considered to establish the generation costs of thermoelectric units, it is subtracted from the social cost to avoid any double counting. Table 9 reports all cost assumptions of the aforementioned externalities as assumed for the presented analysis [64].

**Table 9.** Cost of pollutant emissions assumed to calculate the environmental impact of power system dispatching.

|  | Social Cost | ETS Permits' Cost | Residual Cost of Externality | Unit of Measurement |
|---|---|---|---|---|
| $CO_2$ | 100 | 95 | 5 | EUR/ton |
| $NO_x$ | 39,500 | - | 39,500 | EUR/ton |
| $SO_2$ | 25,400 | - | 25,400 | EUR/ton |
| $PM_{2.5}$ | 100,100 | - | 100,100 | EUR/ton |
| $PM_{10}$ | 38,000 | - | 380,00 | EUR/ton |

Finally, possible benefits of EV aggregates participating in the ASM are assessed with the same market modeling tools presented above. EVs are modeled as limited-energy-content units where charging profiles, as calculated in Section 3.1, are considered as DAM programs, while flexibility profiles (always calculated according to Section 3.1) are used to evaluate upward and downward available power. EVs' scheduled programs and flexibility profiles consider the possible evolution of the EV state-of-charge during an ASM call for each charging event. Therefore, EVs are aggregated based on the market zone and the charging mode, thus obtaining 63 virtual power plants (VPPs) able to provide both power reserves and real-time balancing services. The bids presented by these EV aggregates are set at 115 EUR/MWh for upward regulation and 55 EUR/MWh for downward regulation, which are competitive and close to DAM prices [65]. The roundtrip efficiency of EV batteries is assumed to be 80%. A minimum service duration of one hour is considered for the requested regulating power provision; this is assumed for all storage technologies and for both FRR and RR. Moreover, the available FRR power band is limited to 15% of the total available flexibility band.

VGI benefits are hence estimated by comparing the reference power system dispatching scenario (dumb EV charging, NO VGI scenario) with a scenario where the EVs participate in the balancing market (VGI scenario). In particular, the following aspects are evaluated: ASM cost reduction, as the difference between ASM costs in the NO VGI and VGI scenarios; overgeneration reduction, quantified in terms of avoided NP-RES curtail-

ment in the VGI scenario; EVs' actual participation in power system balancing, assessed in terms of balancing energy volumes provided by EV aggregates and the power reserves procured by exploiting EVs' flexibility; and environmental benefits, calculated considering the variation in pollutant emissions between the NO VGI and VGI scenarios.

While most of the results are presented for the accelerated case (coherent with FF55 policies), thus investigating the most critical situation in terms of EV diffusion, the base case is used to facilitate an understanding of possible non-linearities in the obtained results when a different EV penetration is considered.

## 3. Results

### 3.1. Charging Demand and Flexibility Profiles

EV charging and flexibility profiles for each charging mode are shown in Figure 8 considering the overall Italian power system in a working day of the cold season. As can be seen, there are predominant charging modes, such as the residential and workplace-related ones. In addition, the weight of the LCVs is not negligible; in fact, this is considerable for both the power demand and the flexibility provided. For passenger cars, it is possible to highlight a correlation between the flexibility available and the average stop duration: residential, workplace, and B2C parking modes present an average stop of longer than 6 h and allow their charging power to be modulated down to zero. However, if the stop is shorter, there is less flexibility available. Fast public charging presents stops generally much shorter than 1 h, except for cases of a lunch or dinner stopover; because of this, flexibility can be provided only at mealtimes. Meanwhile, for heavier vehicles, higher charging powers and a wider availability of V2G increase the flexibility available for short stops.

The overall EV charging profile, obtained by summing the charging profiles for all charging modes, is shown in Figure 9. The different natures of daily and nightly charging modes can be seen. Residential charging and commercial vehicle charging push the evening peak and maintain a relevant charge during night. Then, the morning peak mainly involves workplace and public charging. Considering this average profile, the expected peak power withdrawal in 2030 for EV charging is around 3.1 GW against a power system peak of 60 GW (around 5%). The estimated overall energy demand for EV charging in the accelerated scenario is 15.5 TWh/year, representing 4.2% of the expected power system demand at 2030 (266 TWh/year).

### 3.2. Impact of EVs' Dumb Charging

#### 3.2.1. Impact on Distribution Network Management and Planning

The impact on the DN was estimated by comparing the 2022 and 2030 situations concerning EVs' diffusion. Three different aspects were evaluated: the maximum load factor of each network component, calculated as the maximum power level experienced by the component divided by its rated power; the power profile of each component, presenting the number and duration of violations; and the voltage profile of each component, reporting the number and direction (under- or over-voltages) of violations. By simulating the twelve typical days, it was possible to determine the load factor of network components in both urban and rural grids. Figures 10 and 11 show that great EV diffusion increases the maximum load factor of many network components, especially for LV lines and MV/LV transformers.

The impact on DN components mainly depends on two aspects: the perimeter of grid served by the component, and the charging modes associated with the connected charging stations. LV lines and MV/LV transformers serve a limited spatial perimeter, usually a few hundred meters long. Consequently, they can serve a dozen slow-charging points or a few fast-charging stations. In the first case, the network components can experience violation only when they serve a significant number of slow-charging points; this condition arises only with a high penetration or a non-uniform distribution of EVs (e.g., when EVs are concentrated in certain areas). On the other hand, a small number of fast-charging points can cause violations due to their high charging power. This can be seen

in Figures 10 and 11: even if EVs' diffusion implies a generalized increase in the maximum load factor, overloading exceeds 110% only along LV lines and MV/LV transformers.

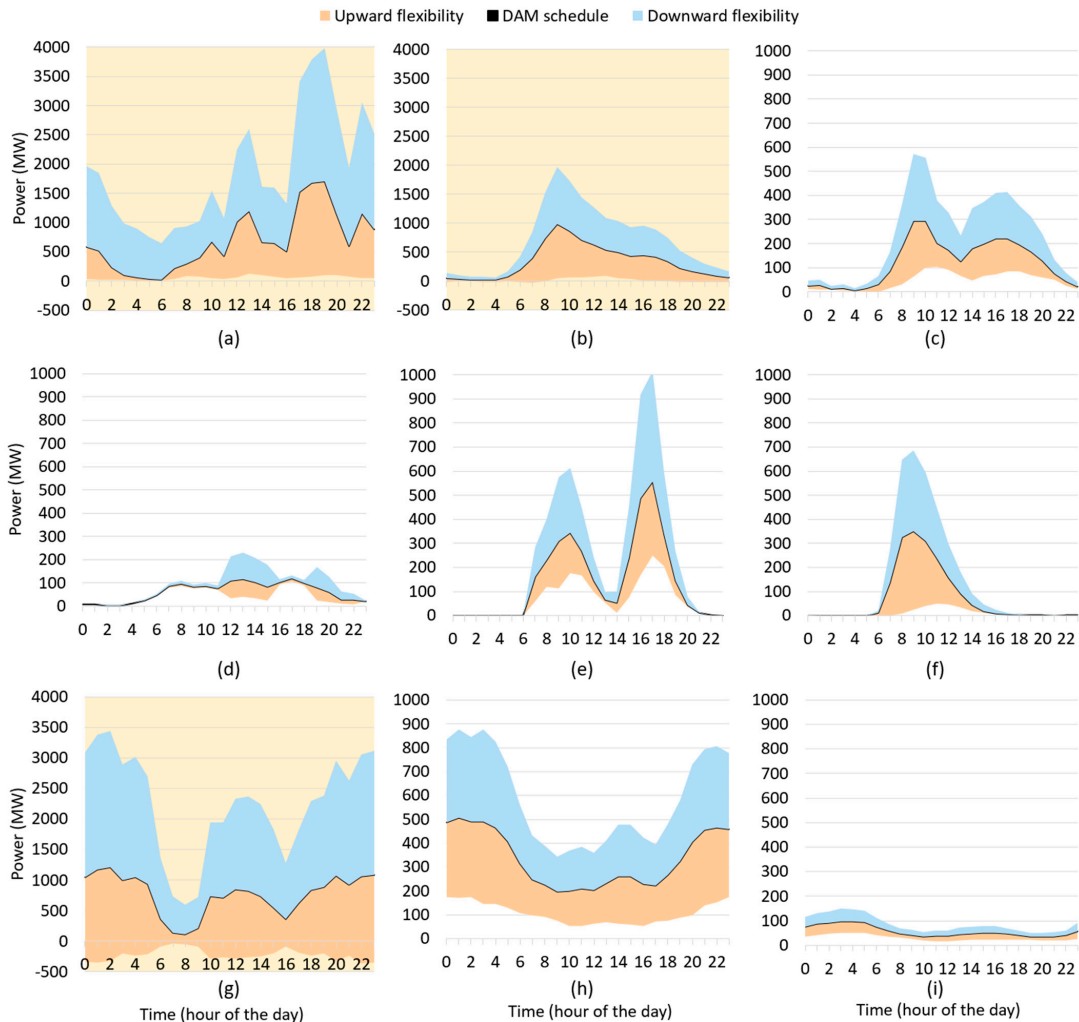

**Figure 8.** DAM (black line) and flexibility (blue and orange areas) profiles in the accelerated case for each charging mode on a working, winter day: (**a**) residential, (**b**) work, (**c**) public slow, (**d**) public fast, (**e**) B2C mall, (**f**) B2C parking, (**g**) LCV, (**h**) HCV, (**i**) PT. Profiles shaded in yellow have a different (larger) scale on the *y*-axis.

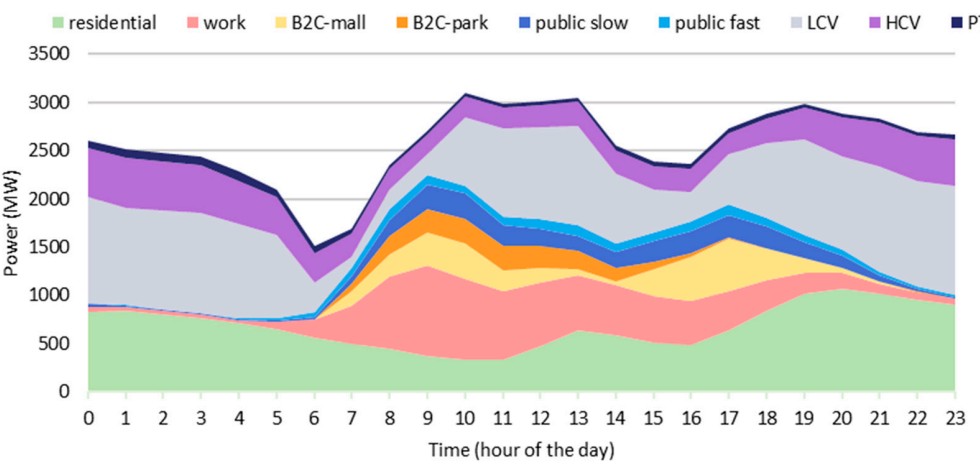

**Figure 9.** Power demand of EVs at the Italian system level on a working, winter day.

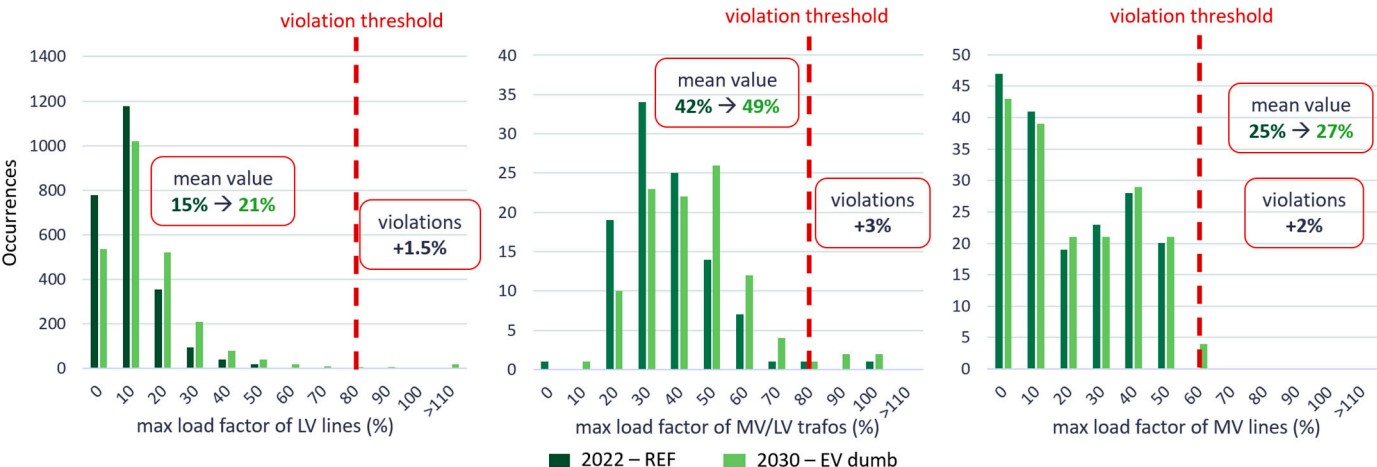

**Figure 10.** Evolution of load factors in the distribution network of rural areas as-is (2022, dark green) and to-be (2030, bright green).

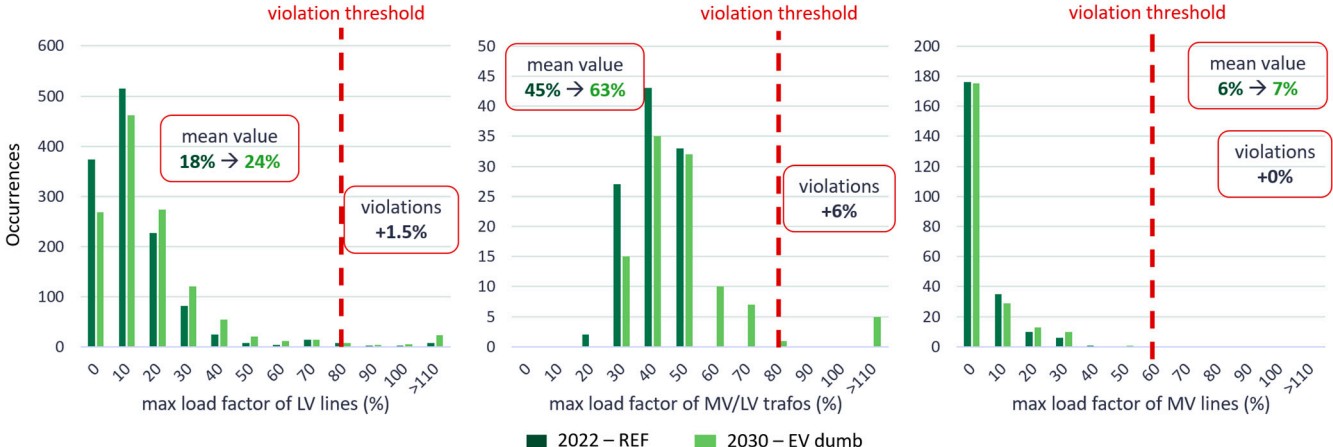

**Figure 11.** Evolution of load factors in the distribution network of metropolitan areas as-is (2022, dark green) and to-be (2030, bright green).

Since EVs usually need to charge a limited amount of energy and the aim is for them to do so in the shortest possible time (i.e., with a high power rate), the probability that many vehicles will charge at the same time is low. Consequently, power violations on low-voltage components are relevant but they have a limited duration, as shown in Figure 12. Moreover, for LV lines, the violation event seems to be independent of the base load (in dark green), meaning that the peak charging power is beyond the design of the DN component. The impact of fast-charging stations becomes less relevant for MV lines and HV/MV transformers: thanks to the wider perimeter of aggregation, the simultaneity factor (i.e., the ratio between currently charging EVs and total EVs) is lower, and the shapes of power profiles are more regular. Therefore, the number of violations is usually lower than for the other network components. On HV/MV transformers, EV charging increases the peak power withdrawal by roughly 6%, meaning that critical conditions arise only when charging occurs during high-demand periods, i.e., the evening or early morning. In these cases, the violation characteristics in terms of intensity and duration depend on the shape of the existing power demand profile.

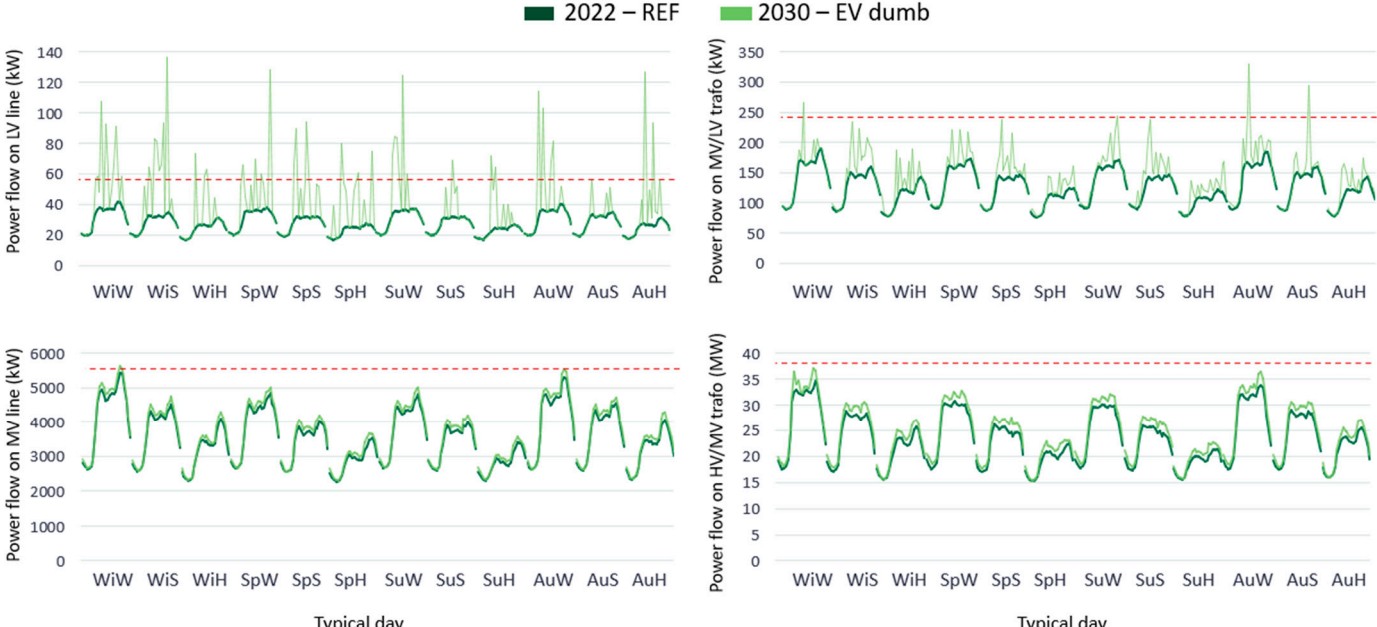

**Figure 12.** Evolution of power profiles for different grid components in a metropolitan network, in the reference year (dark green line) and in a 2030 scenario (light green line). The red dotted line indicates the violation threshold for each network component.

While overload violations show a similar behavior in both rural and urban networks, differences emerge for these networks' voltage profiles, as reported in Figure 13. First, rural networks are characterized by long feeders, increasing the risk of supply voltage variations. Second, rural networks are also characterized by a high penetration of DG, mainly photovoltaic generation; this causes frequent over-voltage events during the day-time. The latter can be avoided by balancing the local power production with the EV charging demand during the daytime. Third, since the nominal power of rural network components is relatively low, a moderate load concentration (e.g., a single fast-charging station) can cause localized voltage issues, especially during the evening or nighttime, temporarily leading to very low voltages.

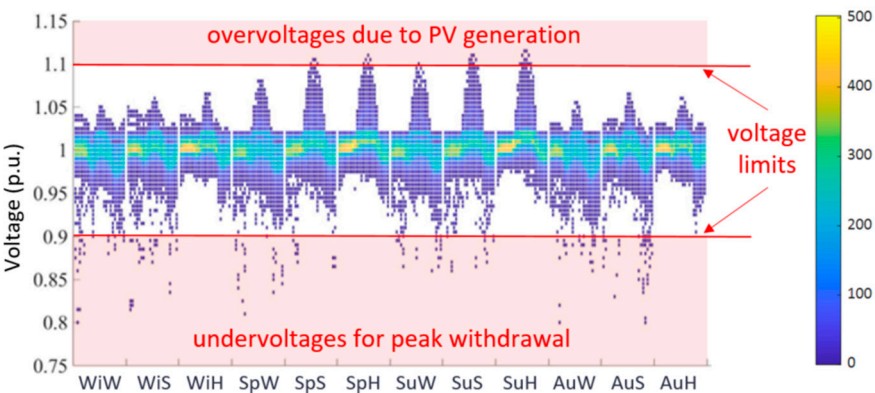

**Figure 13.** Voltage profiles and violations in rural networks. All the voltages in the network are considered. The color represents the frequency in hours per year of that voltage level (blue is the minimum, yellow is the maximum).

### 3.2.2. Impact on Power System Dispatching

EV charging represents a new energy demand for the power system, increasing the uncertainty associated with the load profile forecast and consequently the requested reserve margins and balancing power. Figure 14 presents the results of dumb EV charging, where

EVs do not participate in the ASM, in terms of dispatching costs for the SO (in M EUR) and NP-RES overgeneration linked to power system dispatching (in TWh). For both indicators, base-case and accelerated scenarios show similar results, with around 1.9 B EUR/year in dispatching costs and 5.6 TWh/year of RES curtailed during the ASM activity. This suggests that a greater EV diffusion does not have a major impact on power system dispatching in either economic or technical terms; indeed, dispatching costs are mainly related to the high uncertainty linked to NP-RES generation.

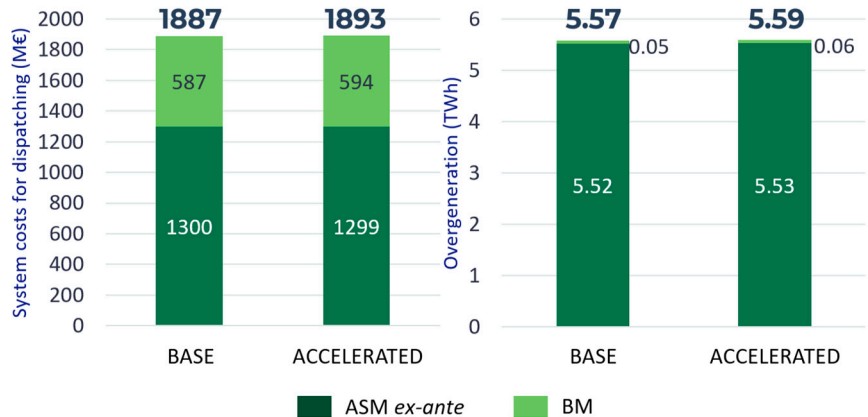

**Figure 14.** Costs and NP-RES overgeneration associated with power system dispatching in the NO VGI case.

As shown in Figure 14, the ASM ex ante phase accounts for almost 70% of the total dispatching costs (1.3 B EUR/year), suggesting that power reserves' provision represents a fundamental issue for the SO when seeking to ensure power system security. The remaining 0.6 B EUR is associated with the balancing market phase (BM), where real-time imbalances are solved. Figure 14 also shows that RES overgeneration emerging on the ASM is around 5.5 TWh in both EV penetration scenarios, almost entirely associated with the ASM ex ante phase; this means that the upward and downward power reserves needed to run the system safely may require turning on some thermoelectric plants, replacing the generation quota from RESs, which will eventually result in overgeneration being curtailed.

### 3.3. Benefits to the Implementation of VGI Solutions

### 3.3.1. Benefits of VGI Solutions for Distribution Network Management and Planning

The identified VGI strategies for the distribution network aim at solving different issues emerging from dumb charging simulations. Evening and morning peaks are reduced through smart control of residential and workplace charging processes, respectively. The latter additionally implies better matching with local photovoltaic generation. Furthermore, the installation of batteries reduces peak power withdrawals associated with fast-charging stations. Finally, local photovoltaic production allows for reducing the issues related to morning peaks of workplace and public charging and supports storage systems in their peak shaving activity. Table 10 reports the total storage and photovoltaic capacity installed in each network archetype, as resulting from the simulations.

**Table 10.** Total installed energy storage and PV capacity resulting from VGI implementation.

| Network | Storage (MWh) | Photovoltaic (MW) |
| --- | --- | --- |
| Urban | 3.3 | 3.0 |
| Rural | 0.8 | 0.8 |

VGI strategies were applied with an additive approach. This considers the realistic and progressive implementation of VGI solutions in the DN alongside great EV diffusion:

capital-light strategies are firstly adopted (smart charging), while physical assets (BESS and PV) are introduced afterward.

The benefits of each VGI practice vary depending on the charging mode and network component. Figure 15 shows the implementation of smart charging based on the V1G solution: having a controlled charging power leads to a smoother and less spiky power profile, while ensuring the EV user's energy demand is met. In the case of residential charging, smart charging almost halves the evening peak while increasing the power withdrawal from midnight to 8 AM. Smart charging is also beneficial for MV/LV transformers and MV lines since it prevents the overlap of EVs' absorption with the evening peak of the residential load. Meanwhile, there is less of a benefit on LV lines since they can be affected by local phenomena: for example, the larger withdrawal in the morning could overlap with high-power public charging. Instead, on LV lines, the control of workplace charging points reduces violations, especially by decreasing the overlap between the non-domestic load and public charging stations.

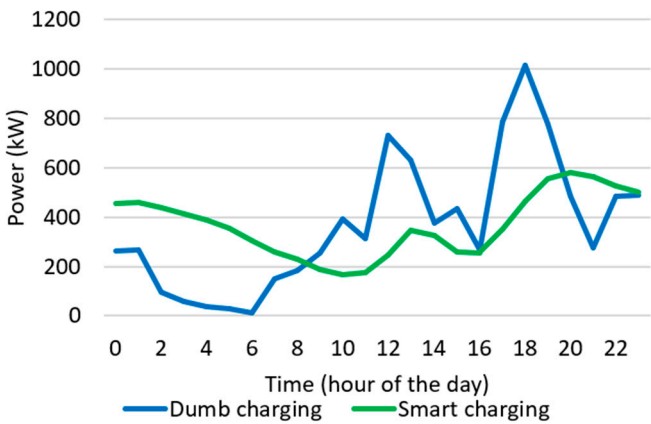

**Figure 15.** Power profiles for dumb (blue) and smart (green) residential charging.

Furthermore, the installation of storage systems is particularly beneficial for LV lines, cutting the short-duration, high-intensity power peaks induced by fast charging, while it is less relevant for other network components, which are less affected by high-power charging. Finally, the installation of power plants brings an overall improvement to the urban distribution network, especially regarding the volume of violating energy. Meanwhile, it has less of an impact on the rural network, where PV generation is already high, and it can sometimes even worsen the performance of the rural network.

Simulations for the DN showed that the benefits of VGI solutions can be fully exploited only when different techniques are applied together. Combining different VGI solutions applied to diverse charging modes can effectively reduce the impact of EVs' diffusion on DN violations. Figures 16 and 17 present the results for the working conditions of both urban and rural network components, obtained by implementing all the considered VGI solutions additively.

In the urban network, the violation number is halved, and the reduction in the violating energy volumes is particularly significant. In the rural network, the benefits are even greater, with almost all the violations solved. Still, photovoltaic generation reduces the number of components experiencing a violation, while increasing the violating energy volume on LV lines. This is because additional PV generation is added in some cases where there is already high PV penetration; consequently, better results could be achieved if the adoption of VGI solutions was somehow coordinated or driven by the DSO, which could indicate the best nodes where PV would be mostly beneficial. This could be achieved in different ways: an example is the so-called Renewable Energy Communities (RECs), which favor spatial and temporal coordination between PV production and energy consumption.

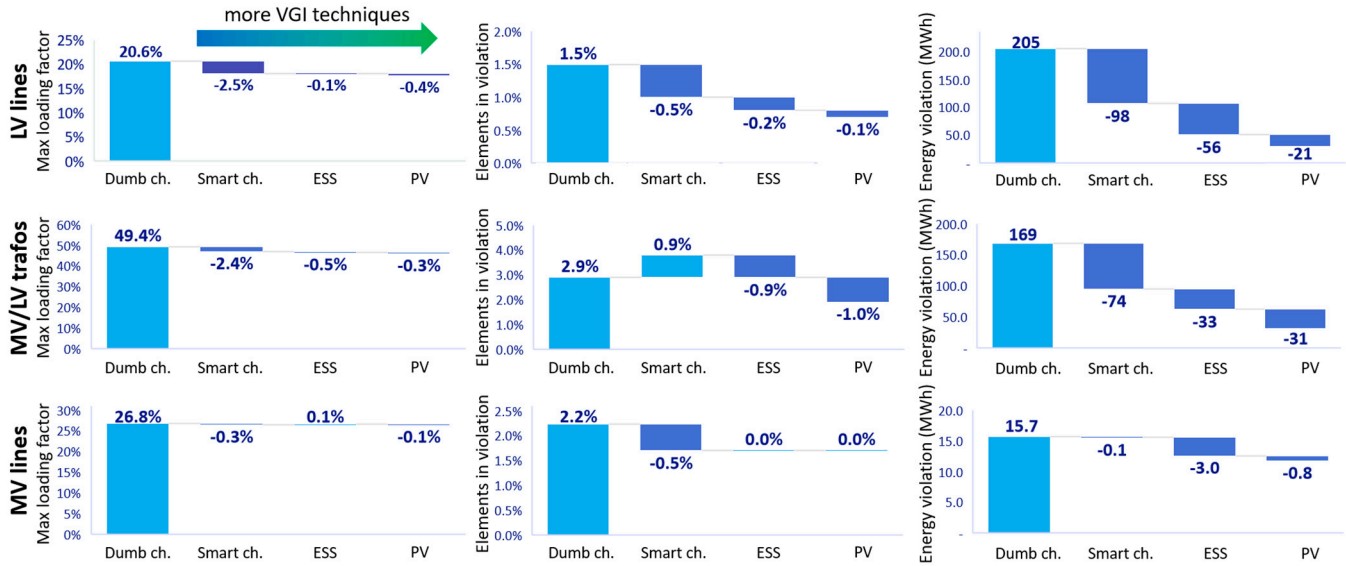

**Figure 16.** Impact of VGI implementation on LV lines (**top**), MV/LV transformers (**mid**), and MV lines (**bottom**) for the distribution network of an urban area.

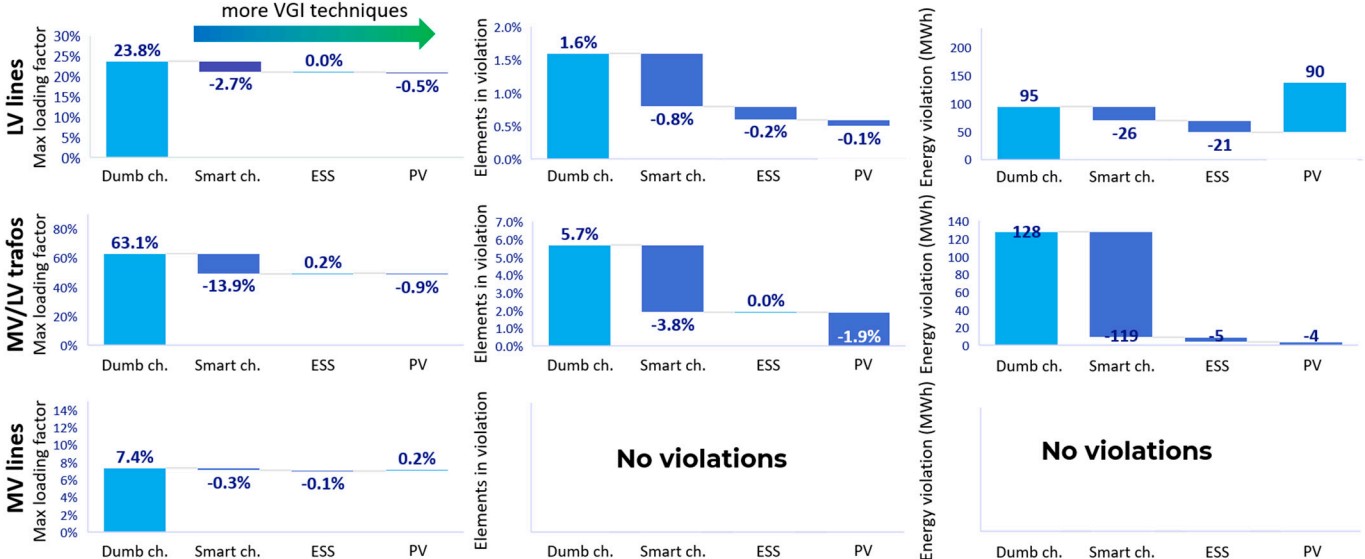

**Figure 17.** Impact of VGI implementation on LV lines (**top**), MV/LV transformers (**mid**), and MV lines (**bottom**) for the distribution network of a rural area. Lightblue is for increase terms, dark blue is for reduction terms.

Finally, the improvement of voltage quality thanks to VGI solutions in rural networks is presented in Figure 18. The improvement is obtained through the implementation of smart-charging techniques and PV-EV coordination in the daytime. Unsolved issues, still remaining after VGI practices' implementation, should be solved by means of other interventions, such as network reinforcement.

### 3.3.2. Benefits of VGI Solutions for Power System Dispatching

While simulations of VGI implementation at a power system level were carried out for both the base-case and accelerated scenarios, only results for the latter are reported in detail since the outcomes in the two scenarios were qualitatively similar and the accelerated scenario was the most stressful for the power system. Figure 19 compares both dispatching costs and ASM-related overgeneration in the NO VGI and VGI scenarios; the results

highlight a 43% (−810 M EUR/year) reduction in the power system dispatching cots and a 47.5% (−2.62 TWh/year) decrease in NP-RES curtailment on the ASM when passing from the NO VGI to the VGI case.

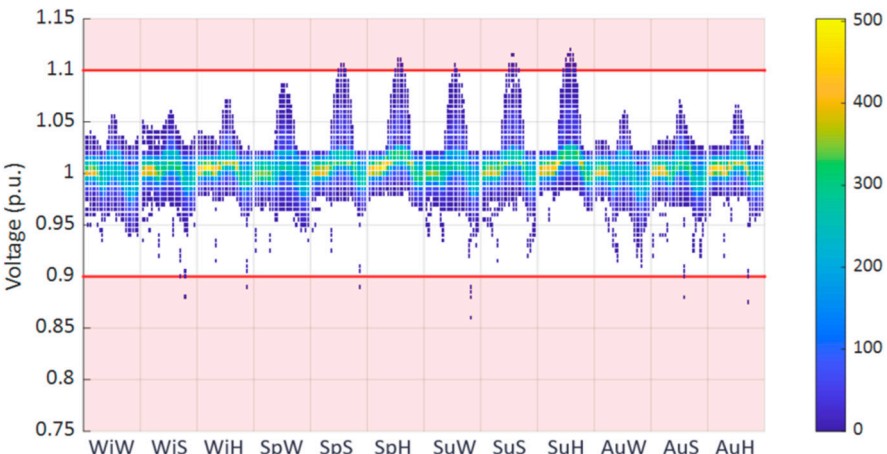

**Figure 18.** Voltage profiles and violations in rural networks with VGI. The color represents the frequency in hours per year of that voltage level (blue is the minimum, yellow is the maximum).

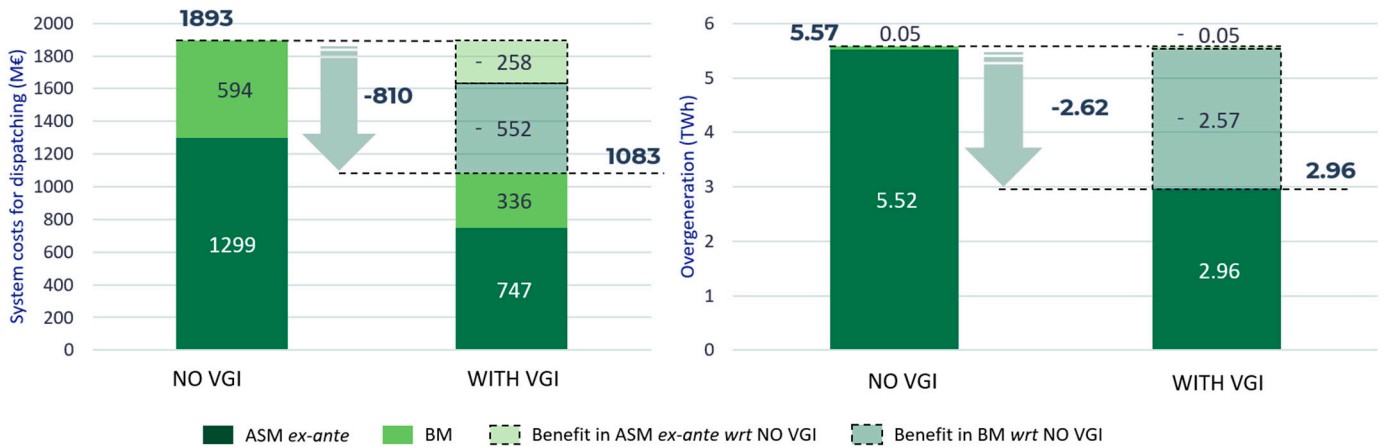

**Figure 19.** Dispatching costs and NP-RES overgeneration comparison for NO VGI vs. VGI case.

Many savings are associated with the ASM ex ante phase when the SO is creating the necessary power-regulating reserve to maintain system security. In the NO VGI situation, the SO is obliged to interrupt thermoelectric and storage plants' DAM schedule to create the required reserve, resulting in high costs; instead, EVs are often naturally available to provide the necessary reserve, without the need to change their charging schedule. This implies an important cost reduction for the SO, which does not have to intervene in the ASM ex ante phase with upward and downward calls. During the balancing phase, cost saving is related to the price of using EVs for their upward and downward flexibility, which is more convenient than that of thermoelectric and storage units most of the time. This is acceptable when considering that the flexibility profile calculation is constrained by the need to take a no harm approach toward the final EV user. Efforts to balance cost reduction, as reported in Figure 19, already include EVs' remuneration for their regulating energy. However, for the ASM ex ante phase, it is instead worth noting that in Italy, no balancing capacity remuneration is foreseen: ex ante costs to procure the required reserve are thus energy-only payments.

A further benefit obtained in the VGI condition is the reduction in ASM-related overgeneration: the downward reserve margins made available from EVs connected in the middle of the day remove the need to switch on thermoelectric units that would

eventually cause RES curtailment. In addition, EVs' flexibility can be shifted toward the hours around midday, thus integrating a larger amount of NP-RES production into the power system balance.

In detail, the system dispatching results show that EVs cover 6% of upward and 21% of downward reserve margins, with an overall weighted contribution to the power reserve of 15% (Figure 20, left). The upward energy volume required during the ASM ex ante phase is around 5 TWh (Figure 20, middle), 2.5 TWh lower than in the NO VGI scenario, since EVs' participation in the ASM often removes the need to switch on thermoelectric units. During the balancing phase, EVs are widely activated both for upward and downward regulation, with a share of 63% of the total activated regulating energy (Figure 20, right).

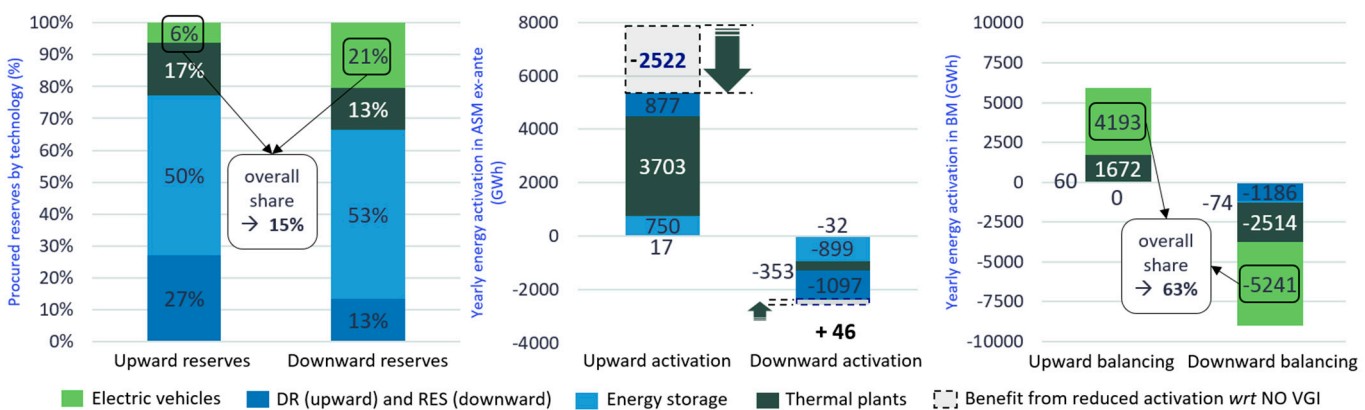

**Figure 20.** Reserve procurement, energy activated in ASM scheduling phase, and balancing in WITH VGI case, accelerated scenario.

Flexibility is mainly provided by charging characterized by a long-duration stop or a small volume of charging energy requested. The daily profiles of reserve margins procured by EVs, calculated as the yearly summation of reserve margins for the same hour of the day, are shown in Figure 21, where the warm and cold seasons are distinguished. The downward reserve provision is mainly characterized by the contributions from residential charging and LCVs, which are the modes where most of the energy is charged (see Figure 9). Meanwhile, the upward reserve provision sees good contributions from workplace and B2C charging modes, especially in the middle of the day. The yearly sum of the 24 h profiles of EVs' procured reserves presents a maximum in the early morning for the upward reserve, while it is in the early afternoon for the downward one, when PV generation is at its maximum. For the same reason, the upward reserve from EVs is greater in the cold season than in the warm one; vice versa, the downward reserve is greater in the warm season.

For the balancing phase, Figure 22 shows the upward and downward balancing energy from EVs for each charging mode, summed over the simulated year. It can be noted that downward energy volumes peak in the early afternoon, when electric vehicles are used to absorb the increased production from photovoltaic sources.

EV participation in the ASM generates both economic and environmental benefits, which arise since these vehicles largely displace thermoelectric power plants. In the presented simulation, EV revenues result from an offered price assumption equal to 115 EUR/MWh for upward bids and 55 EUR/MWh for downward bids. These prices are close to DAM prices (i.e., the difference between the upward ASM offered price and average DAM price is around 10 EUR/MWh) and better than the prices for using other resources. In July 2023, the difference between upward ASM ex ante and DAM average prices was 81 EUR/MWh, and it was 70 EUR/MWh for the balancing phase prices [66]. To understand the impact of a possible variation in the bidding strategy, a sensitivity analysis was performed considering the possibility of strategic behavior by EV managers. Two price scenarios were compared to the reference one (REF): a mid- and a high-price scenario. Figure 23 presents the results in terms of both system benefits (reduction in dispatching

costs) and EV revenues. When bidding at mid prices, the revenues for the EV aggregator increase and the SO cost saving is reduced. This means that the flexibility provided by EVs is better remunerated (higher prices), while, at the same time, the volumes remain constant or slightly decrease, since the offered price is still better than those of other technologies. In the opposite case, the high-price scenario induces a reduction in both system benefits and EV revenues, due to a drop in EVs' flexibility volumes that are activated; this means that EVs are now less convenient than other resources, and thus they are not exploited by the SO. To avoid strategic EV behavior and maximize the flexibility provided, capacity-based remuneration could be implemented besides the energy-only payment currently foreseen in the Italian model.

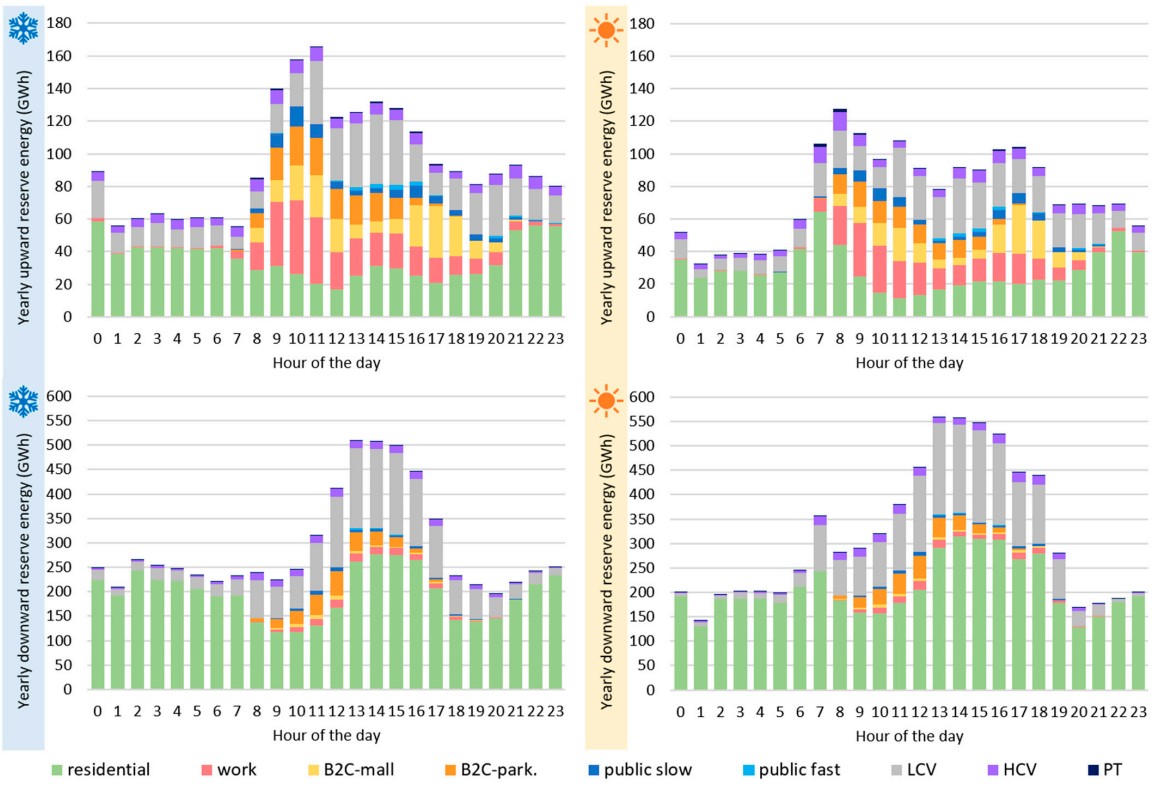

**Figure 21.** Yearly reserve provided by each charging mode in cold (**left**) and warm (**right**) seasons, represented on a 24 h time profile.

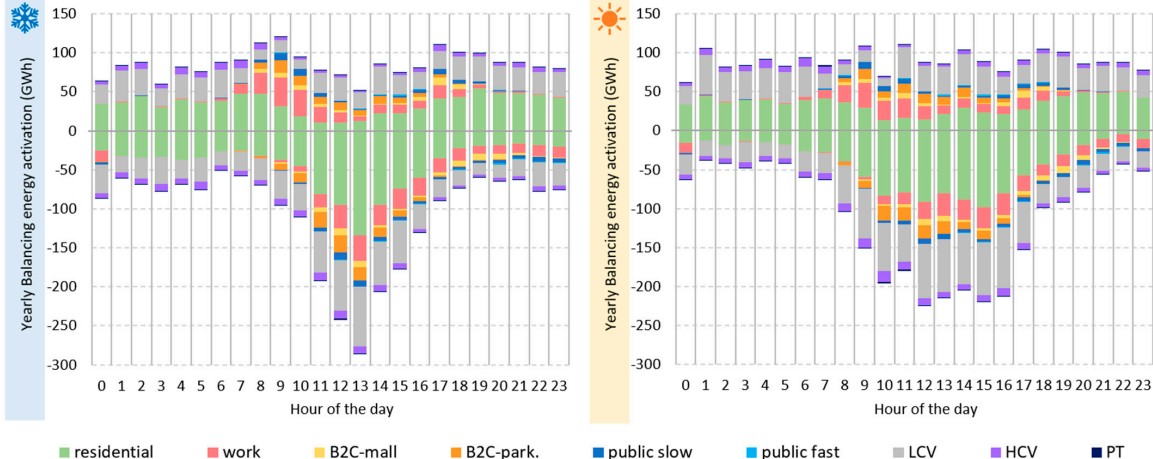

**Figure 22.** Yearly balancing energy provided by each charging mode in cold (**left**) and warm (**right**) seasons, positive for upward (discharging) and negative for downward (charging).

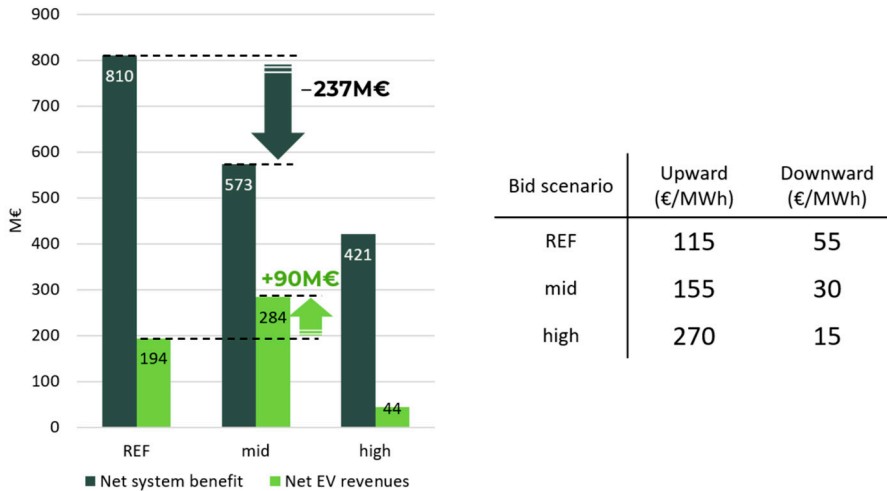

**Figure 23.** Sensitivity analysis results (**left**) and inputs (**right**).

Emissions avoided thanks to EVs are monetized considering the social costs of GHGs and pollutants' externalities. Detailed emissions of greenhouse gases (carbon dioxide, nitrogen oxides, sulfur dioxide) and particulates (PM 2.5 and PM 10) for NO VGI and VGI scenarios are reported in Table 11. The resulting environmental benefit is equal to 34 B EUR per year.

**Table 11.** Pollutants and social costs.

| Product | Emissions—NO VGI (ton) | Emissions—VGI (ton) | Avoided Cost—VGI (B EUR) |
|---|---|---|---|
| $CO_2$ | 1,633,000 | 958,000 | 3.4 |
| $NO_x$ | 1473 | 864 | 24.1 |
| $SO_2$ | 75 | 44 | 0.4 |
| PM 2.5 | 118 | 69 | 4.9 |
| PM 10 | 41 | 24 | 1.2 |

As a general final remark, we highlight that the analysis of the base case versus the accelerated case showed generally linear behavior: the impacts of EVs and the benefits of VGI were linearly proportional to the EVs deployed, exhibiting no breaking point.

## 4. Discussion

This paper has evaluated the impact of EVs' diffusion on the Italian power system and quantified the benefits of vehicle–grid integration for both DN development costs and power system dispatching expenses. The first step consisted of formulating a hypothesis for the Italian 2030 scenario, concerning both power system consistency and EVs' diffusion in the country. A detailed framework was developed to classify and characterize the different EV charging modes, including passenger cars and goods and public transportation. Following this, EV power charging profiles, together with the corresponding upward and downward flexibility available, were calculated through a Monte Carlo procedure that simulated a single EV charging episode; the outcome was a specific hourly charging and flexibility profile, defined over a daily horizon, for each charging mode within every market zone. This distinguished between working days and holidays, warm and cold days, and metropolitan and rural areas. Therefore, it was possible to estimate the impact of EV charging on both DNs and power system dispatching costs. For the former, a Monte Carlo procedure simulating the EV charging process on 12 different typical days was applied on archetypes of metropolitan and rural DNs. The outcomes were the expected active and reactive power flows along the DNs, which allowed for estimations of both overloading and voltage fluctuations. For the latter, a market simulation tool, including both the day-ahead and BM clearing, was used to assess the unit commitment, the power

system constraints, the exchange of flexibility services, and the final dispatching costs. The tool has an hourly resolution, thus allowing for a detailed computation of all power system dispatching variables. Finally, VGI practices were included in both DN and TN management, consisting of the possibility to control the charging power (V1G and V2G), the utilization of a BESS coupled with power charging stations, and the maximization of the local NP-RES self-production for EV charging purposes.

Table 12 summarizes the most important impacts of EV charging emerging from the simulation, distinguishing between DN and power system dispatches. Low-voltage lines suffer because of spatial and temporal clustering of EV charging episodes, which causes violations with a short duration but great intensity. Meanwhile, the overlapping of vehicles' charging and the base load, especially in the evening, causes violations with a lower magnitude but a longer duration along medium-voltage lines. In general, urban DNs demonstrate problems mainly due to lines' overloading, while rural networks present criticalities related to voltage fluctuations because of the asynchronous operations of PV plants (over-voltages) and the load (under-voltages), the latter comprising EV charging. Meanwhile, when we consider power system dispatching, the impacts of EV charging on the total energy demand and system peak load are negligible (4% and 5%, respectively). The main issue is the great uncertainty that EV charging introduces with respect to forecasting the power demand profile; however, while implying a need for greater power reserves, the resulting impact on dispatching costs is negligible.

**Table 12.** Summary of the most important impacts of EVs' mass diffusion on distribution network development costs and power system dispatching expenses.

| | |
|---|---|
| Impacts on distribution network development costs | Violations with short duration (some minutes) but high intensity on LV lines, well distributed during the daytime and mostly linked to fast charging<br>Violations with good duration (>30 min) and low intensity on MV lines, mainly due to EV charging and base load overlapping during the evening<br>Urban areas characterized by overloading phenomena (short lines + high load density), while rural areas interested by voltage fluctuations (long lines + major PV penetration) |
| Impacts on power system dispatching expenses | Poor impact on total system demand (+4%) and peak load (+5%)<br>Negligible impact on dispatching costs because the uncertainty linked to EV charging is much lower than NP-RES production |

The results of the analysis conducted showed that power grids can greatly benefit from VGI. Table 13 reports the main outcomes of the simulations for both DNs and power system dispatching.

**Table 13.** Summary of the most important benefits of VGI implementation for distribution network planning and power system dispatching costs.

| | |
|---|---|
| Benefits for distribution network development costs | Smart-charging (V1G and V2G) practices reduce the network load factor by 13% on average, with a specific advantage during morning and evening load peaks |
| | BESSs coupled with fast and ultra-fast charging reduce the number of overloading violations, especially on low-voltage lines<br>The coordinated exploitation of NP-RES production for EV charging reduces the overloading and voltage fluctuation issues |
| Benefits for power system dispatching expenses | Enabling EVs to system dispatch avoids the start-up of thermoelectric units and the curtailment of NP-RES during the ASM ex ante phase, reducing the corresponding overgeneration by 2.5 TWh/y (45% of the ASM-related overgeneration)<br>EVs' contribution to system dispatching is relevant both ex ante and in real-time, with 15% of total power reserves allocated to EVs (6% of upward ones, 21% of downward ones), and 9 TWh/y of regulating energy provided by EVs out of a total of 15 TWh/y (4 TWh/y upward and 5 TW/y downward) |

When we consider distribution networks, VGI is linked to three main benefits. First, demand response actions applied to EV charging in DNs, and mainly driven by implicit price signals such as time-of-use (ToU) charging tariffs, allow for a 13% reduction on average in the DN load factor. The implementation of smart-charging procedures results in a general smoothing and flattening of the demand profile, which brings advantages for both the average load factor and the volume of energy during violations. The demand response is particularly important in two situations: in the evening (18:00–21:00), within clusters characterized by the residential load, and during the morning (8:00–11:00), when the penetration of commercial and business users is higher. In addition, the demand response (including both V1G and V2G) is fruitful, especially for long-duration charging, which allows for sufficient flexibility in the management of the charging power to reach the desired SoC. Second, BESSs are useful when coupled with fast charging, resulting in a substantial reduction in the overloading episodes (−30%). Since fast charging is characterized by short-duration stops, it is typically not possible to directly modulate the charging power; the use of a BESS allows for preventively storing part of the energy needed to charge the EV, thus reducing short but very high power peaks. This has a great benefit, especially for low-voltage lines. Moreover, the lower peak power required for fast charging allows charging infrastructures characterized by a high overall power (>100 kW) to connect to low-voltage networks. Finally, the possibility of coordinating EV charging with PV production implies a great benefit for the DN, reducing the energy exchanged during violations by up to 70%, thanks to a reduced frequency and intensity of violation events themselves. This contingency is particularly relevant for metropolitan networks, while the longer distances and smaller load density in rural areas make it less useful.

In terms of power system dispatching, it is possible to individuate technical, financial, and environmental advantages of VGI. First, the participation of EVs in system dispatching almost halves the PV overgeneration associated with the ASM ex ante phase (−2.5 TWh/y). Indeed, the exploitation of the reserve margins provided by EVs unlocks those allocated to NP-RES, thus avoiding RES curtailment arising from the operations of thermoelectric units, which has both economic and environmental benefits. Second, EVs' aggregates play a fundamental role within power system dispatching, providing big reserve volumes (15% of the total) and contributing to power regulation (26%). During the ex ante phase, EVs largely replace thermoelectric and storage units, especially in terms of providing the downward regulation reserve, and during the balancing phase, EVs' contribution is very strong in the middle of the day, when they are used to absorb excess PV production. For this purpose, long-duration charging has specific importance, including both nighttime deposits (private or public) and daytime parking, mainly linked to workplaces and modal exchange hubs. The great contribution of EVs to system dispatching is related to the natural predisposition of power charging profiles to provide power regulation reserves, removing the need to rearrange the units' scheduling for the DAM, as happens with thermoelectric plants. These technical benefits result in an overall saving of 800 M EUR per year linked to dispatching activity, equal to 40% of the reference dispatching cost in 2030 (about 3 B EUR/y). This combines both ex ante and balancing savings, equal to −550 M EUR/y and −250 M EUR/y, respectively. In addition, utilizing EVs has a big environmental advantage thanks to the displacement of thermoelectric units (−1.5 TWh/y of produced energy), with an estimated reduction in social costs of around 30 B EUR/y, including both $CO_2$ and other pollutants.

Leveraging the obtained results just highlighted, further works could be draw together the most important implications, which should drive policy and regulatory actions at an international level to promote the diffusion and exploitation of VGI practices worldwide. These could be taken within a two-layer overarching framework:

- First, the flexibility actions of EV charging managers should be driven by economic signals. In this regard, the EV charging cost can be influenced in two ways: implicit economic signals assume that the final user is subject to a certain charging price structure that pushes them to modify their charging profile, while explicit economic

signals remunerate the final user for the provision of flexibility services that they are available to provide, typically through some market mechanism.

- Second, the activation and usefulness of flexibility actions is strongly correlated with the temporal and spatial dimensions. The former concerns the importance of time in this transfer, while the latter regards the location factor. Both affect how much and how well we can transfer a certain economic signal to the final user.

Examples of implicit price signals with a temporal dimension include time-of-use charging tariffs, while advanced or tailored connection procedures are again implicit signals but with a spatial dimension. Meanwhile, explicit price signals with spatial dimensions include non-firm connections or the promotion of aggregated flexibility; moreover, short-term balancing markets represent a solution in which explicit payments can be made for power flexibility.

While the presented analysis has provided a clear and overarching view of the technical and economic potential of VGI, there are still specific issues that should be addressed and limitations to the applicability of VGI. The most relevant of these concern the boundary conditions, consisting of both the diffusion of EVs and the consistency of the power system. First, EV diffusion is not dependent on these vehicles' potential to act as balancing resources for the power system; instead, it is linked to technical and financial evaluations made by citizens and varying from country to country. As such, a further sensitivity analysis of EV scenarios to understand the impact of a different number and/or a spatial diffusion of circulating EVs would be beneficial. Also, it must be noted that the practical implementation of a flexibility provision implies the use of software and hardware for controlling and monitoring smart charging as well as for communicating with the flexibility market player, the BSP. Additionally, we did not consider frequency regulation (namely, a frequency containment reserve) among the provided services. It was absent from this work because, through interviews with technology providers, we found that this service can be demanding for standard supply equipment. This issue should be addressed in dedicated analyses. In addition, the working operations of the power system are mainly influenced by NP-RES installation and infrastructural development, rather than by EVs' diffusion. Nonetheless, the key advantage of implementing VGI stands in the possibility of utilizing for balancing purposes (and RES integration) something that is there with a different goal: moving people and goods around.

**Author Contributions:** Conceptualization, G.R. and F.B.; methodology, G.R., A.C., G.V. and F.B.; software, G.R., A.C. and G.V.; data curation, A.C. and G.V.; writing—original draft preparation, G.R. and F.B.; writing—review and editing, G.R., A.C., G.V. and F.B.; visualization, G.R.; supervision, F.B. All authors have read and agreed to the published version of the manuscript.

**Funding:** This research received no external funding.

**Institutional Review Board Statement:** Not applicable.

**Informed Consent Statement:** Not applicable.

**Data Availability Statement:** Data will be made available upon request. The data are not publicly available since they are partially proprietary.

**Acknowledgments:** The authors thank the Motus-E association for the fruitful discussions and interactions.

**Conflicts of Interest:** Giacomo Viganò was employed by Ricerca Sul Sistema Energetico (RSE). The remaining authors declare that the research was conducted in the absence of any commercial or financial relationships that could be construed as a potential conflict of interest.

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
