# Peer review of "Assessing the Nationwide Benefits of Vehicle–Grid Integration during Distribution Network Planning and Power System Dispatching"

_wevj, doi:10.3390/wevj15040134_

Round 1

Reviewer 1 Report

Comments and Suggestions for Authors

Clarify how the previous papers handle the constraints during the analysis and design

Clarify the limitations of the practical implementation.

Clarify how the previous papers select the parameters of the proposed methods.

Clarify how the previous papers prove the robustness of the proposed methods against data manipulation and noise.

Comments on the Quality of English Language

None

Author Response

Dear Reviewer,

thank you for your comments. Please find detailed answers here below in blue. We believe that the paper improved after the reviews have been implemented.

Clarify how the previous papers handle the constraints during the analysis and design.

Thank you for the comments. Previous papers handle the constraints in different manners, also coherently with the application and the considered framework. For instance, the choice of selecting V1G and V2G is often related to the charging infrastructure and charging mode considered and/or to the supply equipment involved. Also, the modelling of ancillary services markets is often related to pilot experiences or an assumption on possible future layouts. We believe we have depicted the definition of considered constraints concerning the blocks of our composite model, for instance see Figure 4 for charging constraints and see Figure 7 for unit commitment and electricity market constraints.

Clarify the limitations of the practical implementation.

Thank you for your comment. We added a paragraph to limitations and issues in the final discussion. We highlighted that practical implementation deals with software and hardware for communication, monitoring and control, that we did not consider and that could be studied in a dedicated work.

Clarify how the previous papers select the parameters of the proposed methods.

Thank you for the comment. Due to the various nature of analyzed papers, the selected parameters and criteria were several, depending on the application. We also implemented the following paragraph to better highlight the literature choices when dealing with EV charging studies.

“To the best of the authors’ knowledge, parameters were selected in the literature based on real data or on analyses made on a specific case study and no literature review were carried out, for instance on charging behavior and how to use them for nationwide studies.”

Clarify how the previous papers prove the robustness of the proposed methods against data manipulation and noise.

Thank you for your comment. Indeed, previous papers made large use of real data from pilot projects and other local experiences, that could need a proper strategy against manipulation and noise. Actually, our study focuses on a wide set of references coming from statistical studies, rather than real data (this is the case, for instance, for reserve sizing, for EV charging profiles, for circulating fleet description). Therefore, we believe that this topic is far from the scope of the article. We hope the reviewer agrees.

Reviewer 2 Report

Comments and Suggestions for Authors

The paper focuses on estimating the nationwide benefit of automobile network integration in distribution network planning and power system dispatch. The results of the paper show that the spread of electric vehicles will have localized impacts on power and voltage constraints in the distribution network, while the effects on the transmission and dispatch networks will be negligible in terms of capacity and energy demand. In the 2030 scenarios, smart charging reduces grid element disturbances, dispatch costs and RES curtailment.

However, there are comments on the paper:

1. The Abstract section should be rewritten, reflecting in it the relevance of the problem solved and the scientific novelty of the solution obtained.

2. the Abstract section should be removed from the abbreviations (EVs) and others, moving them to the main text of the paper.

3. Keywords should be corrected by highlighting special terms that characterize the research.

4. In the introduction section, the relevance and novelty of the research being conducted should be outlined.

5. In the list of cited sources, more up-to-date publications on the research of electric cargo vehicles and their power sources should be cited, e.g. https://doi.org/10.3390/math11030536, https://doi.org/10.3390/app12199683

6. The criteria for energy flexibility of the power system should be emphasized in more detail.

7. Have and how have energy losses been taken into account when using the V2G mode of electric vehicles? 

8. It should be clarified whether parameters such as incomplete (shallow) charging of the electric vehicle to save idle time and consequently more charging acts have been taken into account in the energy dispatcher system?

9. BESS systems with energy storage are highly costly due to the high cost and degradation of storage device batteries. To what extent has this economic aspect been considered in modeling the benefits of electric vehicle charging in power system dispatch?

10. The Results and Discussion section should further characterize the obtained models and the obtained scientific results, describe their properties, advantages and disadvantages, and cite the limitations of the proposed power system.

11. The conclusions should be structured by highlighting the main scientific and especially practical results obtained, as well as recommendations for power system designers.

Author Response

Dear Reviewer,

thank you for your comment, that we have implemented and answered here below in blue. We think that the paper improved after this review.

Best Regards.

However, there are comments on the paper:

  1. The Abstract section should be rewritten, reflecting in it the relevance of the problem solved and the scientific novelty of the solution obtained.
  2. the Abstract section should be removed from the abbreviations (EVs) and others, moving them to the main text of the paper.

Thank you. The abstract have been widely reviewed, both including the novelties and removing the abbreviations.

  1. Keywords should be corrected by highlighting special terms that characterize the research.

We integrated the keywords:

smart charging; vehicle-grid integration; distribution planning; power system dispatching; flexi-bility; reserve margins

  1. In the introduction section, the relevance and novelty of the research being conducted should be outlined.

Thank you. We added a brief paragraph to highlight the identified research gaps. In particular, we found that literature lacks with nationwide and whole-power-system assessments of EV penetration. We do this proposing a study where all available on-grid VGI solutions are considered, thus including smart charging, V1G, V2G, BESS and NP-RES integration. As well, the study considers at the same time both the local and the system level, providing a complete overview on the impacts and benefits for the power system at distribution and transmission networks level. In addition, VGI implementation is considered for a massive possible adoption, by looking at the intrinsic flexibility of all charging modes (no-harm approach). While the study looks at the Italian system scenario at 2030 as a case study, both the implemented methodology and the obtained results are fully generalizable.

  1. In the list of cited sources, more up-to-date publications on the research of electric cargo vehicles and their power sources should be cited, e.g. https://doi.org/10.3390/math11030536, https://doi.org/10.3390/app12199683

Thank you for the comment. The proposed sources delve into the design and performance optimization of cargo transport. We instead aim at considering the forecasted penetration considering top-down or bottom-up scenarios at national/institutional level.

  1. The criteria for energy flexibility of the power system should be emphasized in more detail.

Thank you. We added a further sentence to clarify that the criteria adopted for power system flexibility needs are retrieved from the regulation and grid codes and represent the institutional estimates considering the expected generating mix and load.

  1. Have and how have energy losses been taken into account when using the V2G mode of electric vehicles? 

As highlighted in the article, the energy losses are modeled as battery efficiency: roundtrip efficiency is 80%, thus considering a cautious approach and the general system performance.

  1. It should be clarified whether parameters such as incomplete (shallow) charging of the electric vehicle to save idle time and consequently more charging acts have been taken into account in the energy dispatcher system?

Thank you for the comment, that allowed us to better highlight in the paper that “the proposed SoC range (see table 3) influences the requested energy per charging event. Thus, the possibility of incomplete charging has always been considered. This results in more charging events for the same overall energy demand”

  1. BESS systems with energy storage are highly costly due to the high cost and degradation of storage device batteries. To what extent has this economic aspect been considered in modeling the benefits of electric vehicle charging in power system dispatch?

Thank you. A sentence has been added to clarify that the BESS costs have not been considered. Oppositely to what has been done in the whole work, the analysis is here only technical, concerning the possible support of stationary BESS.

  1. The Results and Discussion section should further characterize the obtained models and the obtained scientific results, describe their properties, advantages and disadvantages, and cite the limitations of the proposed power system.

Thank you, we added some sentences for considering limitations, such as the fact we do not consider hardware and software to control and monitor smart charging. Also, we suggested future works to better link policies with exploitation of flexibility.

  1. The conclusions should be structured by highlighting the main scientific and especially practical results obtained, as well as recommendations for power system designers.

We thank the reviewer for the observation that we decided to implement. Indeed, we inserted a whole paragraph providing a discussion of policy and regulatory implications coming from the presented analysis. In this, we discuss the potential, highlighted from obtained results, of both implicit and explicit price signals for vehicle-grid integration, declining them also with a temporal and a spatial dimension.

Reviewer 3 Report

Comments and Suggestions for Authors

The paper deals with assessing the nationwide benefits of vehicle-grid integration on distribution network planning and power system dispatching in Italy. It explores the impact and benefits of electric vehicle (EV) diffusion on the national power system, estimating demand and flexibility profiles through dedicated models and a Monte Carlo approach to manage uncertainties in EV charging behavior. Overall, Manuscript is well structured with a logical flow from introduction to conclusion, facilitating easy navigation through the sections. However, there are few suggestions to further improve the manuscript. 

  1. It is suggested to implement consumer behavior as well in terms of charging patterns, which vary with age.
  2. The authors are suggested to give a brief idea of how charging time can be optimized by using the different machine learning algorithms.
  3. It is suggested that authors add an impact on the frequency of the grid as well. In cases of high EV penetration, the charging power required at an instant will be very high, reducing the frequency of the grid.
  4. ⁠Authors are suggested to integrate a small section of the cyber threat possibilities due to smart grid infrastructure and what effect it could have on the electric vehicle charging stations.
Comments on the Quality of English Language

Minor editing is required. 

Author Response

Dear reviewer,

Thank you for your positive feedback and your comments. we implemented a paragraph in the conclusions to state, as a limitation of the study, that we did not consider frequency regulation (namely, Frequency Containment Reserve) among provided services. This is because, after some interviews with technology providers, we found that this service can be demanding for standard supply equipment. Further suggestions have not been integrated since we believe that they are far from the scope of this work. E.g., we do not deal with machine learning, but with standard control strategies. Also, we do not consider cyber security and other costs related to VGI, since we are oriented to a system and market assessment. In conclusion, the proposed charging patterns are average statistical patterns, estimated by a thorough literature review to offer a complete coverage of charging events. Clearly, some peculiarities (eg age of drivers) have been lost in the necessary simplifications. We hope you agree with the implementations done and the proposed motivations.

Round 2

Reviewer 1 Report

Comments and Suggestions for Authors

The authors handled the comments

Comments on the Quality of English Language

None

Reviewer 2 Report

Comments and Suggestions for Authors

The authors have finalized the article and the article can be accepted for publication in my opinion.